# OFFLINE EQUILIBRIUM FINDING IN EXTENSIVE-FORM GAMES: DATASETS, METHODS, AND ANALYSIS

## ABSTRACT

Offline reinforcement learning (Offline RL) brings new methods to tackle real-world decision-making problems by leveraging pre-collected datasets. Despite substantial progress in single-agent scenarios, the application of offline learning to multiplayer games remains largely unexplored. Therefore, we introduce a novel paradigm *offline equilibrium finding* (Offline EF) in extensive-form games (EFGs), which aims at computing equilibrium strategies from offline datasets. The primary challenges of offline EF include i) the absence of a comprehensive dataset of EFGs for evaluation; ii) the inherent difficulties in computing an equilibrium strategy solely from an offline dataset, as equilibrium finding requires referencing all potential action profiles; and iii) the impact of dataset quality and completeness on the effectiveness of the derived strategies. To overcome these challenges, we make four main contributions in this work. First, we construct diverse datasets, encompassing a wide range of games, which form the foundation for the offline EF paradigm and serve as a basis for evaluating the performance of offline EF algorithms. Second, we design a novel framework, BOMB, which integrates the behavior cloning technique within a model-based method. BOMB can seamlessly integrate online equilibrium finding algorithms to the offline setting with minimal modifications. Third, we provide a comprehensive theoretical and empirical analysis of our BOMB framework, offering performance guarantees across various offline datasets. Finaly, extensive experiments have been carried out across different games under different offline datasets, and the results not only demonstrate the superiority of our approach compared to traditional offline RL algorithms but also highlight the remarkable efficiency in computing equilibrium strategies offline.

## 1 INTRODUCTION

Extensive-form games (EFGs) provide a versatile framework for modeling the interactions between multiple players under stochastic and imperfect information settings (Nisan et al., 2007). The canonical solution concept is Nash Equilibrium (NE), where no player can increase his own utility by unilaterally deviating. There are various methods designed for solving extensive-form games, including linear programming (Shoham & Leyton-Brown, 2008), double-oracle algorithms (McMahan et al., 2003), counterfactual regret minimization (CFR) (Zinkevich et al., 2007), and policy-space response oracles (PSRO) (Lanctot et al., 2017). These methods have been successfully applied to real-world large-sale EFGs, e.g., pursuit-evasion games (Xue et al., 2021; Li et al., 2023), poker games (Brown & Sandholm, 2018; 2019; Zha et al., 2021) and Stratego (Perolat et al., 2022).

Despite the successes, existing algorithms require continuous interaction with the game environment or an accurate simulator. For example, CFR-based algorithms necessitate traversing the game tree to compute regret values, and PSRO and its variants demand simulations within the game environment to compute the best response oracle and estimate the entries in the meta-game. We call this paradigm to compute NE as "*online equilibrium finding*". However, in many real-world applications, such as sports games (Liu et al., 2022), network intrusion detection (Khraisat et al., 2019), and automated negotiations (Kiruthika et al., 2020), the immediate interaction with the environment may be expensive and inefficient and the accurate simulator cannot be built. Therefore, offline learning is a preferred option for equilibrium finding in real-world applications.

Figure 1: Comparison between online and offline equilibrium finding

Offline reinforcement learning (Offline RL) has successfully tackled numerous real-world problems by leveraging its offline learning paradigm (Levine et al., 2020). These algorithms fall into two categories: model-free and model-based. Model-free approaches, such as Best-Action Imitation Learning (BAIL) (Chen et al., 2020), directly learn optimal policies from datasets. Conversely, model-based approaches, like Model-based Offline Policy Optimization (MOPO) (Yu et al., 2020) first construct a dynamic model from the dataset, then proceed with planning. The success of these algorithms showcases the significant impact of the offline learning paradigm in advancing RL applications. In recent years, there have been several attempts to formalize the offline learning paradigm in the context of games. StarCraft II Unplugged (Mathieu et al., 2021) provides a dataset of human game-plays in a two-player zero-sum symmetric game. Some previous works (Cui & Du, 2022; Zhong et al., 2022) also explore the necessary properties of offline datasets of two-player zero-sum Markov games to successfully infer their NEs. However, these works mainly focus on solving Markov games, leaving a gap in the literature when it comes to solving extensive-form games in the offline setting. Furthermore, to our understanding, there has been no study focusing specifically on multi-player games in an offline setting. More importantly, there is a notable absence of systematic definitions and research efforts aimed at formalizing offline learning within the context of games.

To address this gap, we propose the novel ***offline equilibrium finding*** (Offline EF) paradigm, which computes the equilibrium strategies using offline datasets. There are several challenges for offline EF. First, the absence of comprehensive benchmarking standards complicates the evaluation and comparison of algorithm performance. Without universally accepted benchmarks, it becomes difficult to objectively measure progress within the field. Second, accurately computing or approximating equilibrium strategies solely from offline datasets is inherently difficult. Specifically, data from just two action profiles are often insufficient for determining proximity to an equilibrium strategy, as equilibrium identification requires all other potential action profiles for reference (Cui & Du, 2022). Third, the quality and completeness of data within offline datasets can significantly impact the effectiveness of derived strategies. Offline datasets fail to cover all possible game states, and this lack of comprehensive coverage can skew the algorithm's ability to generalize from the available data.

This work presents a comprehensive investigation of Offline EF. Specifically, our contributions are fourfold: i) We curate a collection of diverse offline datasets, including random datasets, expert datasets, learning datasets, and hybrid datasets in different extensive-form games; ii) we propose the BOMB framework, which integrates behavior cloning and model-based methods along with a novel parameter estimation method and the model-based method can incorporate any online EF algorithm, e.g., CFR, into the offline context; iii) we provide a comprehensive theoretical analysis for our BOMB framework, offering performance guarantees under different datasets; and iv) we demonstrate the effectiveness of our BOMB framework in computing equilibrium strategies offline through extensive experiments on various offline datasets.

## 2 PRELIMINARIES

**Imperfect-Information Extensive-Form Games.** We use a tuple to represent an imperfect-information extensive-form game (IIEFG), i.e., $\mathcal{G} = (N, H, A, P, \mathcal{I}, u)$ (Shoham & Leyton-Brown, 2008). The set of players is represented by $N = \{1, ..., n\}$, and $H$ represents the set of histories (i.e., the possible action sequences). Especially, the root node of the game tree is represented by the empty sequence $\emptyset$, which is included in $H$. Every prefix of a sequence in $H$ is also included in $H$. The set of terminal histories is represented by $Z$ and belongs to $H$, i.e., $Z \subseteq H$. $A(h) = \{a : (h, a) \in H\}$ is the set of available actions at any non-terminal history $h \in H \setminus Z$. $P$ is the player function, which maps each non-terminal history to a player, i.e., $P(h) \mapsto N \cup \{c\}, \forall h \in H \setminus Z$, where $c$ is the

"chance player" representing these stochastic events outside of the players' controls. $\mathcal{I}$ denotes the set of information set, which forms a partition over the set of histories where player $i$ takes actions, such that player $i$ cannot distinguish these histories within the same information set $I_i$. Every information set $I_i \in \mathcal{I}_i$ corresponds to one decision point of player $i$ which means that $P(h_1) = P(h_2)$ and $A(h_1) = A(h_2)$ for any $h_1, h_2 \in I_i$. For convenience, we use $A(I_i)$ and $P(I_i)$ to represent the action set $A(h)$ and the player $P(h)$ for any $h \in I_i$. For each player $i$, a utility function $u_i : Z \to \mathbb{R}$ specifies the payoff of player $i$ for every terminal history. The behavior strategy of player $i$, $\sigma_i$, is a function mapping every information set of player $i$ to a probability distribution over $A(I_i)$, and $\Sigma_i$ is the set of strategies for player $i$. A strategy $\sigma_i$ is defined as a pure strategy if $\forall I_i \in \mathcal{I}$ and $\forall a \in A(I_i)$, $\sigma_i(I_i, a) \in \{0, 1\}$. It is defined as a mixed strategy if $\forall I_i \in \mathcal{I}$ and $\forall a \in A(I_i)$, $\sigma_i(I_i, a) \in [0, 1]$. Moreover, $\sigma_i$ is considered a fully mixed strategy if $\forall I_i \in \mathcal{I}$ and $\forall a \in A(I_i)$, $\sigma_i(I_i, a) > 0$. A strategy profile $\sigma$ is a tuple of strategies, one for each player, $(\sigma_1, \sigma_2, ..., \sigma_n)$, with $\sigma_{-i}$ referring to all the strategies in $\sigma$ except $\sigma_i$. Let $\pi^\sigma(h) = \prod_{i \in N \cup \{c\}} \pi_i^\sigma(h)$ be the reaching probability of history $h$ when all players choose actions according to $\sigma$, where $\pi_i^\sigma(h)$ is the contribution of player $i$ to this probability. Given a strategy profile $\sigma$, the expected value to player $i$ is the sum of expected payoffs of these resulting terminal nodes, $u_i(\sigma) = \sum_{z \in Z} \pi^\sigma(z) u_i(z)$.

**Solution Concepts.** The common solution concept for IIEFGs is Nash equilibrium (NE) (Nash, 1950), where no player can increase their utility by unilaterally deviating. Formally, a strategy profile $\sigma^*$ forms an NE if it satisfies $u_i(\sigma^*) = \max_{\sigma_i' \in \Sigma_i} u_i(\sigma_i', \sigma_{-i}^*), \forall i \in N$. To measure the distance from the NE, we use the metric $\text{NASHCONV}(\sigma) = \sum_{i \in N} \text{NASHCONV}_i(\sigma)$, where $\text{NASHCONV}_i(\sigma) = \max_{\sigma_i'} u_i(\sigma_i', \sigma_{-i}) - u_i(\sigma)$. When $\text{NASHCONV}(\sigma) = 0$, it indicates that $\sigma$ is the NE. Especially, for $n$-player general-sum games, apart from NE, (Coarse) Correlated Equilibrium ((C)CE) is also a common solution concept. Similar to the NE, a CE is a joint mixed strategy in which no player has the incentive to deviate (Aumann, 1987). Formally, let $S_i$ be the strategy space for player $i$ and $S$ be the joint strategy space. The strategy profile $\sigma^*$ forms a CCE if it satisfies for $\forall i \in N, s_i \in S_i, u_i(\sigma^*) \geq u_i(s_i, \sigma_{-i}^*)$ where $\sigma_{-i}^*$ is the marginal distribution of $\sigma^*$ on strategy space $S_{-i}$. Analogous to NE, the (C)CE Gap Sum is adopted to measure the distance from the (C)CE (Marris et al., 2021).

**Why Existing Methods Fail?** Offline RL focuses on learning the optimal strategies in single-agent scenarios (Levine et al., 2020), which fails to compute the equilibrium in games with offline datasets. Opponent modeling (OM) (He et al., 2016) are used to predict the opponents' behavior strategies. However, opponent modeling algorithms aim at computing the best response strategy of one player instead of the equilibrium strategy, and they also need access to the game environment, which is not applica-

| Methods | Work w/o env | Converge to equilibrium |
|---|---|---|
| Offline RL | ✓ | ✗ |
| OM | ✗ | ✗ |
| Online EF | ✗ | ✓ |

Table 1: Issues of Existing Methods.

ble to Offline EF. The widely used equilibrium finding algorithms, including no-regret methods, e.g., CFR (Zinkevich et al., 2007) and empirical game theoretic analysis (EGTA), e.g., PSRO (Lanctot et al., 2017), require the interactions with the game environments or an accurate simulator (termed as "online EF") and cannot be applied to Offline EF. A clear comparison of existing methods is presented in Table 1 and App. B provides a detailed discussion of related methods.

**Problem Statement.** To facilitate the widespread application of game theory, we extend the offline learning framework into the extensive-form games and introduce the *offline equilibrium finding* paradigm, which focuses on learning equilibrium strategy from historical game-playing data.

**Definition 2.1** (Offline EF). Let $\mathcal{D}$ be an offline dataset of an IIEFG $\mathcal{G}$, generated by an unknown behavior strategy profile $\sigma$. The goal of the *offline equilibrium finding* paradigm is to deduce a strategy profile $\widehat{\sigma}$ from $\mathcal{D}$ to achieving a minimal gap from the equilibrium strategy $\sigma^*$. Formally, $\widehat{\sigma} = \arg\min_{\sigma' \in \Sigma} \text{GAP}(\sigma', \sigma^*)$, where $\text{GAP}(\cdot)$ is a metric function that measures the gap between a given strategy and the equilibrium strategy. $\sigma$ is an $\epsilon$-equilibrium if $\text{GAP}(\sigma, \sigma^*) \leq \epsilon$.

Building on the definition of the offline EF paradigm, we can instantiate this paradigm by defining a metric for the gap from the equilibrium strategy, such as the $\text{NASHCONV}$ for NE (Nash, 1950) and (C)CE Gap Sum for (C)CE (Aumann, 1987). While offline EF shares similarities with offline RL to some extent, it also presents distinct differences and unique challenges. Firstly, unlike offline RL, which aims to compute an optimal strategy (Levine et al., 2020), the offline EF paradigm seeks to

achieve an equilibrium strategy. This objective necessitates an iterative process to calculate the best response strategy, introducing distinct complexities. Secondly, the offline EF paradigm involves at least two players, making the game dynamics particularly sensitive to distribution shifts and other uncertainties – a stark contrast to offline RL. Thirdly, while in offline RL, the data from two actions may suffice to determine which action is better, in the offline EF paradigm, simply comparing the data of two action tuples is inadequate for identifying which tuple is closer to an equilibrium strategy, as equilibrium identification requires other action tuples for references (Cui & Du, 2022).

## 3 DATASETS

Datasets play a pivotal role in offline learning, however, there are no publicly available datasets specifically tailored for the offline EF paradigm. Consequently, we outline our methods to collect datasets at different expert levels that will serve as a basis for advancing offline EF research.

**Formats.** Before delving into the methods of dataset collection, it is essential to outline the data formats of the offline EF dataset for IIEFGs. The offline dataset can be represented by $\mathcal{D} = (s_t, a_t, s_{t+1}, u_{t+1}, d_{t+1})$. Here, $s_t$ and $s_{t+1}$ represent the game states at time step $t$ and $t+1$ respectively from the game-level perspective. Specifically, $s_t$ encompasses all relevant game information at time step $t$, which includes the information sets for each player and other game information $GI$ out of the control of players, such as the results of

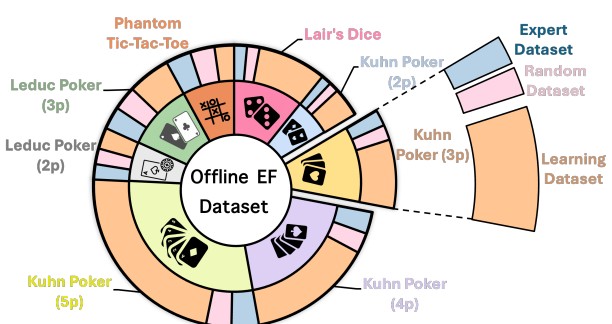

Figure 2: Dataset of Offline EF.

chance node $(I_1^t, I_2^t, ..., I_n^t, GI)$, the player who needs to act $(p^t)$, the set of available actions for the acting player $(A(I_{p^t}^t))$, i.e., $s_t = (I_1^t, I_2^t, ..., I_n^t, GI, p^t, A(I_{p^t}^t))$. Notable, $p^t$ may represent a chance player $c$ to include the game's stochastic events outside of all players' control. The utility for each player at time step $t+1$ is represented by $u_{t+1} = (u_1^{t+1}, u_2^{t+1}, ..., u_n^{t+1})$. Finally, the variable $d_{t+1}$ indicates whether the game ends at state $s_{t+1}$, with a value of 1 if the game ends and 0 otherwise.

**Collecting Methods.** Similar to the practices in the offline RL domain, datasets in the offline EF area must be diverse to serve as effective benchmarks for developing and evaluating algorithms. Many benchmarks in the offline RL area, such as those discussed in (Fujimoto et al., 2019; Gulcehre et al., 2020), collected data from online RL training runs. Additionally, D4RL (Fu et al., 2020) incorporates a range of dataset collection methods inspired by real-world applications, including human demonstrations, exploratory agents, and hand-coded controllers. Inspired by these benchmarks in the offline RL area, we propose several methods for collecting offline datasets at different expert levels. The first one, referred to as the *random method*, involves each player adopting a uniform strategy and participating repeatedly in the game to collect data as the random dataset. This approach is motivated by the innate exploratory tendency and mimics a novice's initial gaming experience. The second method, the *learning-based method*, draws inspiration from the player skill improvement process. We implement an existing equilibrium finding algorithm, such as CFR (Zinkevich et al., 2007) or PSRO (Lanctot et al., 2017), collecting and storing intermediate game interactions to compile the learning dataset. The final method, the *expert method*, capitalizes on insights gained from observing expert players' strategies. In this approach, each player follows an assigned equilibrium strategy and repeatedly engages in the game to generate the expert dataset. Additionally, to enhance realism and increase dataset diversity, we propose a hybrid approach that combines the random and expert datasets in varying proportions, resulting in a more comprehensive collection of datasets.

**Statistics of Datasets.** We developed a benchmark dataset for offline EF, employing previously outlined collection methods on **eight** commonly used IIEFGs, as depicted in Fig. 2. In total, our offline EF dataset comprises approximately **3.8 million** data points, occupying about **11GB** of memory. For each game, we have generated three distinct types of datasets: Expert, random, and Learning, each reflecting our data collection methods. The proportions of each dataset are visually detailed and comprehensive statistics on the distribution of these datasets are detailed further in App. C.2.

# 4 BOMB: FRAMEWORK AND THEORETICAL ANALYSIS

Inspiring by the success of Offline RL, there are two main directions to develop the algorithmic framework for Offline EF: i) behavior cloning (BC) (Fujimoto & Gu, 2021), which basically imitates the strategies used to collect the offline data with additional exploration, and ii) model-based methods (Yu et al., 2020; Kidambi et al., 2020), which first learns a world model from the offline dataset and then learn the strategy from the world model. However, BC may fail when the collecting policy is a random policy, which can be exploited by the opponent and model-based methods may fail when the collecting strategies are an equilibrium strategy, in which only a small portion of the game state is visited. To mitigate these issues, we propose the BOMB framework which combines **B**ehavior cl**O**ning and **M**odel-**B**ased method for offline EF paradigm.

## 4.1 BOMB FRAMEWORK

**BOMB.** Alg. 1 shows the whole framework of BOMB. Given an offline dataset $\mathcal{D}$, we first train the policy $\sigma_\theta$ based on the dataset $\mathcal{D}$ using a behavior cloning (BC) technique (Line 2). Note $\mathcal{D} = (s_t, a_t, s_{t+1}, u_{t+1}, d_{t+1})$ and $s_t = (I_1^t, I_2^t, ..., I_n^t, GI, p^t, A(I_{p^t}^t))$. Since the policy network $\sigma_\theta$ is trained to mimic the behavior strategy, only the information set $I_{p^t}^t$ and the corresponding action $a_t$

---

**Algorithm 1** BOMB Framework

1: **Input:** an offline dataset $\mathcal{D}$
2: Train policy $\sigma_\theta$ based on $\mathcal{D}$ using BC technique;
3: Train an environment model $E_{\theta_e}$ based on $\mathcal{D}$;
4: Learn $\sigma_{mb}$ policy using any EF algorithm on $E_{\theta_e}$;
5: Select $\alpha$ using parameter estimation method;
6: $\sigma = \alpha \cdot \sigma_\theta + (1 - \alpha) \cdot \sigma_{mb}$;
7: **Output:** Policy $\sigma$

---

in $\mathcal{D}$ are required for training. The cross-entropy loss is taken as the training loss, defined as $\mathcal{L}_{bc} = -\mathbb{E}_{(I_{p^t}^t, a_t) \sim \mathcal{D}}[a_t \cdot \log(\sigma(I_{p^t}^t; \theta))]$. On the other hand, inspired by model-based offline RL algorithms, where a dynamic model is trained to simulate the real environment (Kidambi et al., 2020; Yu et al., 2020; Matsushima et al., 2020), we learn an environment model $E_{\theta_e}$ is trained based on dataset $\mathcal{D}$ and $E_{\theta_e}$ is used for learning the MB policy $\sigma_{mb}$ by any online EF algorithm, e.g., PSRO (Lanctot et al., 2017), (Lines 3-4). Specifically, we use the game state $s_t$ and corresponding action $a_t$ as inputs, with the subsequent game state $s_{t+1}$, reward $u_{t+1}$ and the termination variable $d_{t+1}$ serving as labels. Stochastic gradient descent (SGD) is employed as the optimizer for parameter updates, and the mean squared error loss is used as the training loss, defined as $\mathcal{L}_{env} = \mathbb{E}_{(s_t, a_t, s_{t+1}, u_{t+1}, d_{t+1}) \sim \mathcal{D}}[\mathbf{MSE}((s_{t+1}, u_{t+1}, d_{t+1}), E(s_t, a_t; \theta_e))]$. The final policy is obtained to combine the BC and MB policies, i.e., $\sigma = \alpha \sigma_\theta + (1 - \alpha)\sigma_{mb}$ where $\alpha$ denote the weight of the BC policy (Lines 5-6). The estimation method for determining $\alpha$ is introduced below.

**Estimation of Parameter $\alpha$.** Here, we introduce three estimation methods of parameter $\alpha$. The simplest method is randomly selecting a value from the interval $[0, 1]$ as the parameter $\alpha$. Although this method can be implemented fully offline, it lacks guarantees for achieving the most effective combined strategy. The second method is the grid search method, in which we define a set of 11 candidate values for $\alpha$, i.e., $\alpha = \{0, 0.1, ..., 1\}$, and these values are used to configure combined policies, which are then tested in a real environment. The value of $\alpha$ that results in the smallest gap from the equilibrium strategy is selected as optimal. This method can yield the best performance and similar techniques that determine offline parameters or fine-tune offline policies

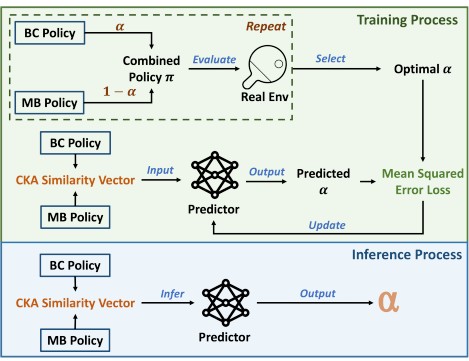

Figure 3: Learning-based estimation method.

through online interactions are commonly employed in offline RL (Kalashnikov et al., 2018; Lee et al., 2022). To render our approach fully offline while still achieving optimal parameter values, we propose a learning-based method, depicted in Fig. 3. In this method, a predictor is trained to estimate $\alpha$ based on the difference between the BC and the MB policies. We first use the grid search method to get optimal parameter values as labels. The predictor takes the centered kernel alignment (CKA) (Kornblith et al., 2019) similarity vector between the BC and the MB policies as input and

outputs the estimated $\alpha$. The predictor can be trained in one game where access to the environment is feasible, and reuse the predictor in similar games. Though the predictor can only provide an approximate optimal parameter value, it requires no further online interactions once trained.

**Advantages of BOMB.** There are several advantages of BOMB. First, by combining the BC and MB, BOMB can work on the datasets collected by any strategies, i.e., either random or equilibrium strategies. Second, with the learned world model for games, BOMB can seamlessly integrate the online EF algorithms, thus BOMB can generalize to different equilibria. For example, for computing NE, we adapt PSRO (Lanctot et al., 2017) and Deep CFR (Brown et al., 2019) methods, referred to as MB-PSRO and MB-CFR, respectively. Additionally, we adapt the JPSRO method (Marris et al., 2021) (MB-JPSRO) for computing (C)CE. iii) BOMB is game-agnostic, which can learn the game rules from the offline datasets and do not rely on the knowledge of the game, which shares the similar advantages with MuZero (Schrittwieser et al., 2020)

### 4.2 THEORETICAL ANALYSIS

In the offline RL area, dataset coverage over the optimal policy is sufficient for offline learning (Rashidinejad et al., 2021; Xie et al., 2021). However, we found the dataset assumption that the dataset generated by the equilibrium strategy is not sufficient for computing equilibrium strategies in an offline manner. It can be confirmed by the counter-example illustrated in Fig. 4. In this game, we can easily get NE strategy, $\sigma^* = (\sigma_1^*, \sigma_2^*) = (\{I_1 : a_1\}, \{I_2 : b_2\})$. If we use this equilibrium strategy to generate the offline dataset $\mathcal{D}$, then $\mathcal{D}$ would only include the data point $((I_1^{t_1} = I_1, I_2^{t_1} = \emptyset, GI = \emptyset, 1, \{a_1, a_2\}), a_1, (I_1^{t_2} = I_1 a_1, I_2^{t_2} = \emptyset, GI = \emptyset, -1, \emptyset), (0, 0), 1)$. Clearly, the dataset $\mathcal{D}$ is not sufficient for computing the NE strategy since there is no information about Player 2. Another assumption — that the equilibrium strategy is covered by the offline dataset — is also insufficient for the offline EF paradigm, as we prove in App. D.1. In this section, we outline the necessary and sufficient conditions for the coverage of an offline dataset that guarantees the convergence of our methods in IIEFGs with perfect recall. We start by introducing two key concepts of dataset coverage: uniform coverage and equilibrium coverage.

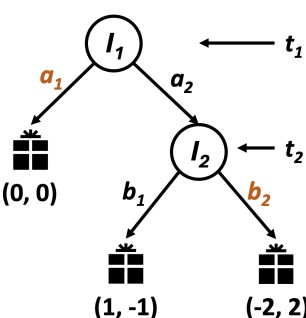

Figure 4: Game example.

**Definition 4.1.** An offline dataset $\mathcal{D}$ is said to be a *uniform coverage* of an IIEFG $\mathcal{G}$ if and only if the offline dataset $\mathcal{D}$ covers all possible state-action pairs. Formally, $(s_t, a_t, s_{t+1}, u_{t+1}, d_{t+1}), \forall s_t, a_t \in A(s_t)$ and $s_{t+1} \in T(s_t, a_t)$ where $T$ is the transition function of game $\mathcal{G}$.

**Definition 4.2.** An offline dataset $\mathcal{D}$ is said to be an $\epsilon$-*equilibrium coverage* over an IIEFG $\mathcal{G}$ if and only if its underlying behavior strategy $\sigma_{\mathcal{D}}$ satisfies $\text{GAP}(\sigma_{\mathcal{D}}, \sigma^*) < \epsilon$, where $\sigma_{\mathcal{D}}$ is defined as $\sigma_{\mathcal{D}}(s_t, a_t) = \frac{C(s_t, a_t)}{C(s_t)}$ and $\sigma_D(s_t, a_t) > 0$ for all $s_t$ and $a_t \in A(s_t)$, with $C(s_t, a_t)$ and $C(s_t)$ denoting the counts of data points containing $(s_t, a_t)$ and $s_t$ in $\mathcal{D}$, respectively.

Building on the dataset coverage definitions previously introduced, we now discuss the conditions under which our method achieves convergence. To facilitate this analysis, we introduce an assumption about the error in training neural networks within the algorithm. All subsequent theorems are derived under this assumption unless stated otherwise.

**Assumption 4.3.** The error in training neural networks within our method is assumed to be smaller than an arbitrarily small $\epsilon$, provided that the dataset contains a sufficient amount of data.

To further support this assumption, we provided a general generalization bound for the training error under a dataset with size $m$ in App. D.2. Then we present our result as follows.

**Theorem 4.4.** *Let $\sigma_{MB(\mathcal{D})}$ be the strategy profile learned by our **model-based algorithm** based on the offline dataset $\mathcal{D}$ with sufficient data under Assumption 4.3. Then, $\sigma_{MB(\mathcal{D})}$ is guaranteed to be an $\epsilon$-equilibrium strategy of the IIEFG $\mathcal{G}$ if and only if $\mathcal{D}$ is a uniform coverage of $\mathcal{G}$ and $\sigma_{MB(\mathcal{D})}$ is an $\epsilon$-equilibrium strategy for the trained environment model within the model-based algorithm.*

*Sketch Proof.* According to Assumption 4.3, the error in training the environment game model based on $\mathcal{D}$ can be considered negligible. Consequently, the trained environment game model is identical

to the original game $\mathcal{G}$, as the dataset $\mathcal{D}$ provides full coverage of all state transitions. Therefore, if $\sigma_{MB(\mathcal{D})}$ is an $\epsilon$-equilibrium strategy for the trained environment game model, it is also an $\epsilon$-equilibrium strategy for the original game $\mathcal{G}$. Any slight violation of these conditions would invalidate the convergence result. A complete proof is provided in App. D.1. □

**Theorem 4.5.** *Let $\sigma_{BC(\mathcal{D})}$ be the strategy profile learned by our **behavior cloning algorithm** based on the offline dataset $\mathcal{D}$ with sufficient data under Assumption 4.3. Then $\sigma_{BC(\mathcal{D})}$ is guaranteed to be an $\epsilon$-equilibrium strategy of IIEFG $\mathcal{G}$ if and only if the offline dataset $\mathcal{D}$ is an $\epsilon$-equilibrium coverage of the IIEFG $\mathcal{G}$.*

*Sketch Proof.* According to Assumption 4.3, the error in training the behavior cloning strategy $\sigma_{BC(\mathcal{D})}$ from the dataset $\mathcal{D}$ is negligible. Therefore, by the behavior cloning process, $\sigma_{BC(\mathcal{D})}$ is identical to the behavior strategy underlying $\mathcal{D}$, i.e., $\sigma_{BC(\mathcal{D})} = \sigma_{\mathcal{D}}$. Consequently, if $\mathcal{D}$ is an $\epsilon$-equilibrium coverage of $\mathcal{G}$, then $\sigma_{BC(\mathcal{D})}$ is an $\epsilon$-equilibrium strategy for the IIEFG $\mathcal{G}$, as $\text{GAP}(\sigma_{\mathcal{D}}, \sigma^*) < \epsilon$ implies $\text{GAP}(\sigma_{BC(\mathcal{D})}, \sigma^*) < \epsilon$. Any slight violation of these conditions would invalidate the convergence result. The full proof is provided in App. D.1. □

Building on the insights provided by the preceding two theorems, we propose the following theorem concerning the performance of BOMB under a general case where an unknown strategy profile generates the offline dataset. The full proof can be found in App. D.1.

**Theorem 4.6.** *Let $\sigma_{BOMB(\mathcal{D})}$ represent the strategy profile learned by our **BOMB algorithm** based on the offline dataset $\mathcal{D}$ with sufficient data under Assumption 4.3, $\sigma_{\mathcal{D}}$ represent the underlying behavior strategy of $\mathcal{D}$ and $\sigma^*$ represent the equilibrium strategy of IIEFG $\mathcal{G}$. Then the gap between $\sigma_{BOMB(\mathcal{D})}$ and $\sigma^*$ is at most equal to, or smaller than, the gap between $\sigma_{\mathcal{D}}$ and $\sigma^*$, i.e., $\text{GAP}(\sigma_{BOMB(\mathcal{D})}, \sigma^*) \leq \text{GAP}(\sigma_{\mathcal{D}}, \sigma^*)$.*

To better analyze the performance of our algorithm under real-world cases, we first analyze the offline dataset we generated for the offline EF paradigm. Based on dataset collection procedures, we find that the random dataset can be considered as a uniform coverage of the game $\mathcal{G}$ when the dataset is sufficiently large. This is because the random dataset is collected using a uniform strategy, ensuring that every action is adequately sampled as long as enough data is collected. On the other hand, the expert dataset can be considered as an $\epsilon$-equilibrium coverage of game $\mathcal{G}$, where $\epsilon$ decreases as the dataset size increases. Since the expert dataset is generated by an equilibrium strategy, a larger sample size means the underlying behavior strategy of the dataset more closely approximates the equilibrium strategy, resulting in a smaller $\epsilon$. Therefore, the above properties of our algorithm hold under these two datasets, as shown in the following experimental results.

## 5 EXPERIMENTAL RESULTS

To assess the performance of our proposed algorithm – BOMB, we conduct the following experiments: i) we compare two offline RL algorithms to our BOMB algorithm; ii) we evaluate the performance of different estimation methods; and iii) we run the BOMB framework on various offline datasets to evaluate its performance in computing different equilibrium strategies.

We use OpenSpiel[1] (Lanctot et al., 2019) as our experimental platform, as it offers a well-established collection of environments and algorithms for game research, thereby facilitating future replicability. We select several poker games, Liar's Dice and Phantom Tic-Tac-Toe, which are all widely used in previous works (Lisý et al., 2015; Brown et al., 2019). Experiments are conducted on a workstation with a ten-core 3.3GHz Intel i9-9820X CPU and NVIDIA RTX 2080Ti GPU. All results are averaged over three seeds and error bars are also reported. To demonstrate the performance of our algorithm, we present our results by answering the following research questions (RQs).

**RQ1:** *Can the BOMB framework outperform offline RL methods?*

To support this claim that offline RL algorithms are insufficient for the offline EF paradigm, we choose one model-free algorithm–Best-Action Imitation Learning (BAIL) (Chen et al., 2020) and one model-based algorithm–Model-based Offline Policy Optimization (MOPO) (Yu et al., 2020) as

---

[1]https://github.com/deepmind/open_spiel

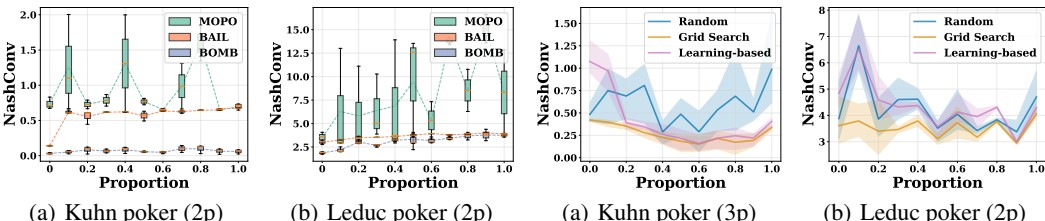

Figure 5: Comparison with offline RL.    Figure 6: Results of different estimation methods.

the representative of offline RL algorithms. Fig. 5 shows the comparison results in two-player Kuhn poker and Leduc poker games under hybrid datasets. The x-axis represents the proportion of data from the random datasets in the hybrid dataset. When the ratio is zero, the hybrid dataset is equivalent to the expert dataset; conversely, when the ratio is one, the hybrid dataset is reduced to the random dataset. We found that BOMB outperforms both offline RL algorithms in all cases. It means that neither of these offline RL algorithms can produce a strategy profile close enough to the equilibrium strategy, which might be attributed to the players' policies being optimized independently.

**RQ2:** *How do different parameter estimation methods perform?*

As introduced previously, we propose three combination methods: random method, grid search method, and learning-based method. To evaluate the performance of different combination methods, we conduct experiments on poker games. For the learning-based method, we train the parameter predictor on the two-player Kuhn poker game. And then, we test the parameter predictor in other poker games. Fig. 6 shows the performance results of three combination methods on three-player Kuhn poker and two-player Leduc poker games. We can find that the grid search method achieves the best performance and the learning-based method performs similarly to the grid search method on the three-player Kuhn poker while it performs slightly worse on the two-player Leduc poker game. It implies that the performance of the parameter predictor mainly depends on the difference between the test game and the game used to train the predictor. The most interesting result is that the random method performs well in many cases, which means that even a simple combination works well. In the rest of the experiments, we use the grid search method as the parameter estimation method.

**RQ3:** *Can the BOMB framework compute NE?*

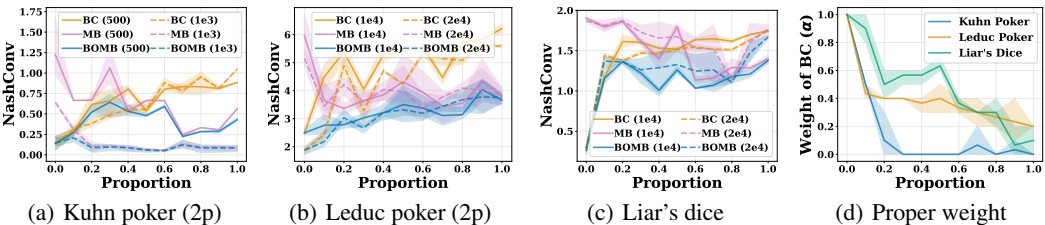

Figure 7: Experimental results on computing NE in two-player games.

To answer this question, we conduct extensive experiments covering two-player cases, multi-player cases, and real-world scenarios simulated using learning datasets. This comprehensive approach allows for an adequate evaluation of our method's performance in computing the NE strategy.

**Two-Player Cases.** We first move to evaluate the performance of our algorithm, BOMB, in computing the NE strategy. In addition to performing the BOMB framework, we also assess the individual performance of the behavior cloning technique and the model-based algorithm. This assessment not only helps in understanding the strengths and weaknesses of each component but also provides a comprehensive insight into the efficacy of the BOMB framework in computing the equilibrium offline. Figs. 7(a)-7(c) show results on some two-player games under different sizes of offline datasets. Here, we use MB-CFR or MB-PSRO to compute the NE strategy. The MB framework's perfor-

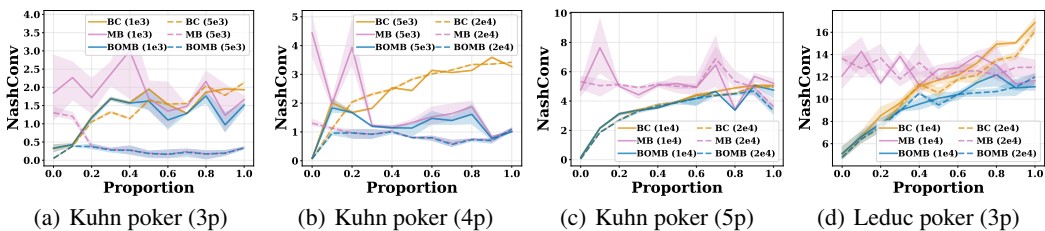

Figure 8: Experimental results on computing NE in multi-player games.

mance is independent of the algorithm used to compute the NE strategy (shown in App. F). As the proportion of the random dataset increases, we observe that the performance of BC decreases while the performance of MB slightly increases. Additionally, we notice that as the size of offline data increases, the improvement of the BC's performance is not significant and the MB's performance improves. It means that the performance of BC mainly depends on the quality of datasets, i.e., the quality of the behavior policy generating the dataset, and the performance of MB relies on the similarity between the environment model and the actual environment. These figures show that our algorithm, BOMB, outperforms both BC and MB methods in all cases, demonstrating its effectiveness in computing NE strategy for two-player imperfect-information extensive-form games.

To further analyze the performance of the BOMB method in two-player games, we plot the parameter $\alpha$ for these combined policies, as illustrated in Fig. 7(d). The results show that as the proportion of the random dataset in the hybrid dataset increases, the weight of the BC policy decreases. It confirms that the BC policy performs better under the expert dataset while the MB policy performs better under the random dataset from another side.

**Multi-Player Cases.** We also conduct experiments on multi-player games, specifically evaluating the performance of our method in computing the NE strategy across several multi-player games, as shown in Fig. 8. The results demonstrate that our BOMB framework consistently performs as well as or better than both BC and MB algorithms, similar to the findings in two-player scenarios. Furthermore, as the proportion of the random dataset increases, the performance of BC decreases, while MB shows instability with a slight downward trend. It is important to note that we adopt MB-CFR as the model-based algorithm, and since CFR-based algorithms do not guarantee convergence to the NE strategy in multi-player games, the performance of MB may be affected. Additionally, the performance of the model-based method also relies on the accuracy of the trained environment game model. Consequently, the underperformance of the MB algorithm may be due to either an inadequately trained environment game model or the limitations of the CFR-based algorithm in multi-player settings. Therefore, developing an effective equilibrium-finding algorithm and training an accurate environment game model are both key challenges for offline EF in multi-player imperfect-information extensive-form games.

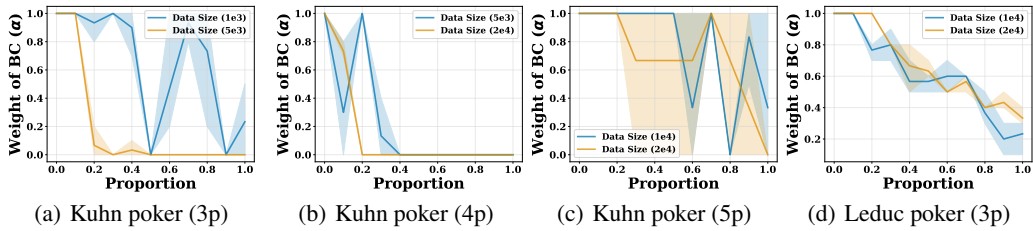

Figure 9: Proper weight of BC policy in multi-player games.

The appropriate weights of the BC policy ($\alpha$) within the BOMB framework across different hybrid datasets are presented in Fig. 9. In three-player and four-player Kuhn poker games, we observe that the weight of the BC policy quickly drops to zero as the proportion of the random dataset in the hybrid dataset increases, indicating that the MB method generally outperforms the BC method, except when the random dataset proportion is low. In contrast, in five-player Kuhn poker and three-

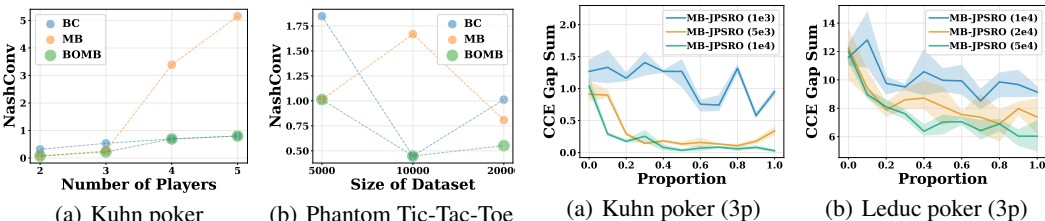

Figure 10: Results on learning dataset.

Figure 11: Results on computing CCE.

player Leduc poker games, the weight of the BC policy remains high in most cases, except when the proportion of the random dataset is high. This may be due to the poor performance of the MB method in these games, highlighting the challenge of learning an approximate equilibrium strategy using the MB method in complex, multi-player games, where both developing an effective equilibrium-finding algorithm and training an accurate environment game model are particularly difficult.

**Simulating Real-World Cases.** We also conduct experiments on the learning dataset, which closely approximates real-world conditions. Fig. 10(a) shows the results of Kuhn poker games with different numbers of players and Fig. 10(b) shows the results of Phantom Tic-Tac-Toe under different numbers of offline data. It indicates that given an offline dataset generated by an unknown strategy, our algorithm can also perform better than BC and MB in approximating the NE strategy.

**RQ4:** *Can the BOMB framework compute CCE?*

We proceed to evaluate the performance of the model-based method in computing the CCE strategy. We do not perform the BC technique and the BOMB framework to compute the CCE strategy since the offline dataset is collected using an independent strategy for each player, rather than a joint strategy. Fig. 11 shows the results of performing the MB-JPSRO algorithm on three-player Kuhn poker and Leduc poker games. We can observe that as the size of the offline data increases, the performance of MB-JPSRO improves. This further supports the notion that the performance of the model-based method primarily depends on the quality of the trained environment model and also highlights its significance in computing equilibrium strategy offline.

## 6 CONCLUSION

We investigated the paradigm of offline equilibrium finding (Offline EF) in extensive-form games, which focuses on finding equilibrium strategies from offline datasets. To be specific, we first created the offline EF datasets using several established data-collecting methods, which solves the challenge of the absence of a comprehensive dataset for evaluation. Then, we proposed a novel algorithm, BOMB, which combines the behavior cloning technique with a model-based approach that can adapt regular online equilibrium finding algorithms to the offline setting by introducing an environment model. To better understand the algorithm, we provide a comprehensive theoretical and empirical analysis, providing performance guarantees of our algorithms across different offline datasets. Finally, extensive experimental results further validated the superiority of the BOMB framework over existing offline RL algorithms, affirming its efficacy for computing equilibrium strategies in an offline manner. We hope our efforts can open up new avenues in equilibrium finding and accelerate research in large-scale game theory.

**Limitations and Future Work.** There are several limitation of this work. First, the games considered are relatively small-scale, and the large-scale games including Texas Hold'em poker (Brown & Sandholm, 2018) and football games (Liu et al., 2022) will be included in the future work. Second, this work primarily focus on NE and CCE, more solution concepts will be considered such as quantal response equilibrium (QRE) (McKelvey & Palfrey, 1995) and $\alpha$-rank (Omidshafiei et al., 2019). We will investigate the genealizability of both the datasets and the BOMB framework to novel solution concepts in the future work. Third, the relationships between the datasets and the offline EF algorithms can be further investigated, where instead of collecting the datasets by researchers, we can apply the offline EF to human-play datasets toward real deployment (Wang et al., 2024).

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

## A   FREQUENTLY ASKED QUESTIONS

**Q1: What are the potential impacts of this work?**

This work fills the lack of offline learning in the game theory field. Benefiting from the offline learning framework, we anticipate that our offline equilibrium finding setting could pave a path to solving real-world problems using these game theory-based methods and inspire new research directions in equilibrium finding. Furthermore, the equilibrium strategy is more robust compared with just the optimal strategy in some security-related scenarios. Consequently, offline EF plays a crucial role in obtaining more robust strategies for tackling these competitive real-world problems.

**Q2: Why offline EF is important and is more difficult than offline cooperative MARL?**

Utilizing offline EF algorithms specifically designed for adversarial environments is crucial in strictly competitive games, such as security games. This setting fundamentally differs from offline multi-agent reinforcement learning, which generally focuses on cooperation between agents rather than strict competition. For instance, consider the class of pursuit-evasion games, where the pursuer (defender) chases the evader (attacker). In this scenario, we cannot make any assumptions about the attacker's strategy beforehand, as the attacker is strategic and capable of learning. Employing a vanilla offline RL algorithm to learn the defender's optimal strategy based solely on historical data might lead to a significant utility loss, as the defender's optimal strategy could be exploitable. In other words, the attacker may switch to the best response against the computed strategy of the defender instead of adhering to their past behavior estimated from the data. Therefore, achieving Nash Equilibrium (NE) may be a more suitable solution, as NE strategies are non-exploitable.

To be more specific, traditional offline RL focuses on learning the optimal strategy, i.e., obtaining the highest utility, for an agent acting in a dynamic environment modeled as a single MDP, which does not depend on the actions of other agents. In contrast, in two-player games, the dynamics for one player depend not only on the environment but also on the strategy of the opponent. In other words, the MDP in which a player acts in games is determined by both the game and the fixed strategy of the opponent, and hence a change in the opponent's strategy instigates a corresponding change in the MDP. This makes computing the best strategy for the defender against a strategic opponent using offline RL significantly more difficult. The framework of offline EF we introduced provides methods for computing a player's NE strategy, which is their optimal strategy against a strategic opponent (i.e., the worst case for the player).

**Q3: What are the differences between Offline EF and EGTA?**

1) As described in (Wellman, 2006), EGTA takes the game simulator as input and performs strategic reasoning through interleaved simulation and game-theoretic analysis. Therefore, the game simulator is required in EGTA. However, only the offline dataset is available in the offline EF paradigm and the game simulator is not required. 2) The estimated game model (empirical game) in EGTA is built based on the simulation's results, which are obtained by performing known strategies on the simulator. In contrast, in the offline EF paradigm, the offline dataset is generated with an unknown strategy. Although we use different behavior strategies to generate several offline datasets, we do not utilize these strategies in the offline EF paradigm.

**Q4: What are the novelties of the proposed Offline EF algorithm – BOMB?**

To our knowledge, we are the first ones to propose an empirical algorithm for computing the equilibrium strategy from the offline dataset, i.e., the offline EF paradigm. Unlike traditional offline RL algorithms, which belong to either model-based or model-free categories, our algorithm combines the advantages of both model-based and model-free approaches to efficiently compute equilibrium strategies in an offline manner. Our BOMB framework integrates the behavior cloning technique with a model-based method, equipping novel parameter estimation methods. We introduce an environment model to design the model-based method that can generalize regular online equilibrium finding algorithms to the offline setting. Furthermore, we proposed several different methods to determine the combination parameter value. In different scenarios, according to whether the online interaction is available, there are corresponding algorithms to determine the parameter value. Finally, experimental results show that BC and MB cannot perform consistently well and BOMB outperforms them in all cases. It indicated that our BOMB framework takes advantage of both algorithms and performs well in computing equilibrium strategies in an offline manner.

## B    MORE RELATED WORK

**Equilibrium Finding Algorithms.**    As described in the main paper, the contemporary state-of-the-art algorithms for solving IIEFGs may be roughly divided into two groups: no-regret methods derived from CFR (Zinkevich et al., 2007), and incremental strategy-space generation methods of the PSRO framework (Lanctot et al., 2017). Next, we will introduce these two classes of algorithms.

For the first group, CFR is a family of iterative methods for approximately solving imperfect-information extensive-form games. Let $\sigma_i^t$ be the strategy used by player $i$ in iteration $t$. We use $u_i(\sigma, h)$ to define the expected utility of player $i$ given that the history $h$ is reached and all players act according to strategy $\sigma$ from that point on. Accordingly, $u_i(\sigma, h \cdot a)$ is used to define the expected utility of player $i$ given that the history $h$ is reached and all players play according to strategy $\sigma$ except player $i$ selects action $a$ in history $h$. Formally, $u_i(\sigma, h) = \sum_{z \in Z} \pi^\sigma(h, z) u_i(z)$ and $u_i(\sigma, h \cdot a) = \sum_{z \in Z} \pi^\sigma(h \cdot a, z) u_i(z)$. The *counterfactual value* of the information set $I$, $v_i^\sigma(I)$, is the expected value of information set $I$ given that player $i$ attempts to reach it. This value is the weighted average of the expected utility of each history in the information set. The weight is proportional to the contribution of all players except player $i$ to reach each history. Thus, $v_i^\sigma(I) = \sum_{h \in I} \pi_{-i}^\sigma(h) u_i(\sigma, h)$. For any action $a \in A(I)$, the counterfactual value of action $a$ is $v_i^\sigma(I, a) = \sum_{h \in I} \pi_{-i}^\sigma(h) u_i(\sigma, h \cdot a)$. The *instantaneous conterfactual regret* for an action $a$ in information set $I$ during iteration $t$ is $r^t(I, a) = v_{P(I)}^{\sigma^t}(I, a) - v_{P(I)}^{\sigma^t}(I)$. Therefore, the conterfactual regret for an action $a$ in inforamtion set $I$ on iteration $T$ is $R^T(I, a) = \sum_{t=1}^T r^t(I, a)$. In vanilla CFR, players use *Regret Matching* to pick a distribution over available actions in an information set proportional to the cumulative regret of those actions. Formally, in iteration $T + 1$, player $i$ selects action $a \in A(I)$ according to probabilities

$$\sigma^{T+1}(I, a) = \begin{cases} \frac{R_+^T(I, a)}{\sum_{b \in A(I)} R_+^T(I, b)} & \text{if } \sum_{b \in A(I)} R_+^T(I, b) > 0, \\ \frac{1}{|A(I)|} & \text{otherwise,} \end{cases}$$

where $R_+^T(I, a) = \max\{R^T(I, a), 0\}$ is the position portion of the regret value since we often are most concerned about the cumulative regret when it is positive. If a player acts according to regret matching in the information set $I$ on every iteration, then in iteration $T$, $R^T(I) \leq \Delta_i \sqrt{|A_i|} \sqrt{T}$ where $\Delta_i = \max_z u_i(z) - \min_z u_i(z)$ is the range of utilities of player $i$. Moreover, $R_i^T \leq \sum_{I \in \mathcal{I}_i} R^T(I) \leq |\mathcal{I}_i| \Delta_i \sqrt{|A_i|} \sqrt{T}$. Therefore, $\lim_{T \to \infty} \frac{R_i^T}{T} = 0$. In two-player zero-sum games, if both players' average regret $\frac{R_i^T}{T} \leq \epsilon$, their average strategies $(\overline{\sigma}_1^T, \overline{\sigma}_2^T)$ over all iterations form a $2\epsilon$-equilibrium (Waugh et al., 2009). Some CFR-based variants are proposed to solve large-scale imperfect-information extensive-form games. Some sampling-based CFR variants (Lanctot et al., 2009; Gibson et al., 2012; Schmid et al., 2019) are proposed to effectively solve large-scale games by traversing a subset of the game tree instead of the whole game tree. With the development of deep learning techniques, neural network function approximation can be applied to the CFR algorithm. Deep CFR (Brown et al., 2019), Single Deep CFR (Steinberger, 2019), and Double Neural CFR (Li et al., 2019) are algorithms using deep neural networks to replace the tabular representation.

For the second group, PSRO is a general framework that scales Double Oracle (DO) (McMahan et al., 2003) to large extensive-form games via using reinforcement learning to compute the best response strategy approximately. To make PSRO more effective in solving large-scale games, Pipeline PSRO (P2SRO) (McAleer et al., 2020) is proposed by parallelizing PSRO with convergence guarantees. Extensive-Form Double Oracle (XDO) (McAleer et al., 2021) is a version of PSRO where the restricted game allows mixing population strategies not only at the root of the game but every information set. It can guarantee to converge to an approximate NE in a number of iterations that are linear in the number of information sets, while PSRO may require a number of iterations exponential in the number of information sets. Neural XDO (NXDO) as a neural version of XDO learns approximate best response strategies through any deep reinforcement learning algorithm. Recently, Anytime Double Oracle (ADO) (McAleer et al., 2022), a tabular double oracle algorithm for two-player zero-sum games is proposed to converge to an NE while decreasing exploitability from one iteration to the next. Anytime PSRO (APSRO) as a version of ADO calculates best responses via reinforcement learning algorithms. Except for NEs, we also consider (Coarse) Correlated equilibrium ((C)CE). Joint Policy Space Response Oracles (JPSRO) (Marris et al., 2021) is proposed for training

agents in n-player, general-sum extensive-form games, which provably converges to (C)CEs. The excellent performance of these equilibrium-finding algorithms depends on the interactions with the actual game environment or a precise simulator. Therefore, these algorithms cannot directly be applied to the offline EF paradigm. In our paper, we propose a model-based method that can adapt existing equilibrium finding algorithms to the offline context.

**Opponent Modeling.** Opponent modeling algorithm is necessary for multi-agent settings where secondary agents with competing goals also adapt their strategies, yet it remains challenging because policies interact with each other and change (He et al., 2016). One simple idea of opponent modeling is to build a model each time a new opponent or group of opponents is encountered (Zheng et al., 2018). However, it is infeasible to learn a model every time. A better approach is to represent an opponent's policy with an embedding vector. Grover et al. (2018) use a neural network as an encoder, taking the trajectory of one agent as input. Imitation learning and contrastive learning are also used to train the encoder. Then, the learned encoder can be combined with reinforcement learning algorithms by feeding the generated representation into the policy and/or value network. DRON (He et al., 2016) and DPIQN (Hong et al., 2017) are two algorithms based on DQN, which use a secondary network that takes observations as input and predicts opponents' actions. However, if the opponents can also learn, these methods become unstable. Therefore, it is necessary to take the learning process of opponents into account. Foerster et al. (2017) propose a method named Learning with Opponent-Learning Awareness (LOLA), in which each agent shapes the anticipated learning of the other agents in the environment. Further, the opponents may still be learning continuously during execution. Therefore, Al-Shedivat et al. (2017) propose a method based on a meta-policy gradient named Mata-MPG. It uses trajectories from current opponents to perform multiple meta-gradient steps and constructs a policy that favors updating the opponents. Meta-MAPG (Kim et al., 2021) extends Mate-MPG by including an additional term that accounts for the impact of the agent's current policy on the future policies of opponents, similar to LOLA. Yu et al. (2021b) propose model-based opponent modeling (MBOM), which employs the environment model to adapt to various opponents. In our offline EF paradigm, our goal is to compute the equilibrium strategy based on the offline dataset. Applying opponent modeling is not enough for the offline EF paradigm since it only aims at computing the best response strategy instead of the equilibrium strategy.

**Empirical Game Theoretic Analysis.** Empirical game theoretic analysis (EGTA) is an empirical methodology that bridges the gap between game theory and simulation for practical strategic reasoning (Wellman, 2006). In EGTA, game models are iteratively extended through a process of generating new strategies based on learning from experience with prior strategies. The strategy exploration problem (Jordan et al., 2010) that how to efficiently assemble an efficient portfolio of policies for EGTA is the most challenging problem. Schvartzman & Wellman (2009b) deploy tabular RL as a best-response oracle in EGTA for strategy generation. They also build the general problem of strategy exploration in EGTA and investigate whether better options exist beyond best-responding to an equilibrium (Schvartzman & Wellman, 2009a). Investigation of strategy exploration was advanced significantly by the introduction of the Policy Space Response Oracle (PSRO) framework (Lanctot et al., 2017) which is a flexible framework for iterative EGTA, where at each iteration, new strategies are generated through reinforcement learning. Note that when employing NE as the meta-strategy solver, PSRO reduces to the double oracle (DO) algorithm (McMahan et al., 2003). In EGTA, a space of strategies is examined through simulation, which means that it needs a simulator, and the policies are known in advance. However, in the offline EF paradigm, only an offline dataset is provided. Therefore, techniques in EGTA cannot be directly applied to the offline EF paradigm.

**Offline Reinforcement Learning.** Offline reinforcement learning (offline RL) is a *data-driven* paradigm that learns exclusively from static datasets of previously collected interactions, making it feasible to extract policies from large and diverse training datasets (Levine et al., 2020). This paradigm can be extremely valuable in settings where online interaction is impractical, either because data collection is expensive or dangerous (e.g., in robotics (Singh et al., 2021), education (Singla et al., 2021), healthcare (Liu et al., 2020), and autonomous driving (Kiran et al., 2022)). Therefore, efficient offline RL algorithms have a much broader range of applications than online RL and are particularly appealing for real-world applications (Prudencio et al., 2022). Due to its attractive characteristics, there have been a lot of recent studies. Here, we can divide the research of offline RL into two categories: model-based algorithm and model-free algorithm.

Model-free offline RL algorithms learn a good policy directly from the offline dataset. To do this, there are two types of algorithms: actor-critic and imitation learning methods. Those actor-critic

algorithms focus on implementing policy regularization and value regularization based on existing reinforcement learning algorithms. Haarnoja et al. (2018) propose soft actor-critic (SAC) by adding an entropy regularization term to the policy gradient objective. This work mainly focuses on policy regularization. For the research of value regularization, an offline RL method named Constrained Q-Learning (CQL) (Kumar et al., 2020) learns a lower bound of the true Q-function by adding value regularization terms to its objective. Another line of model-free offline RL research is imitation learning which mimics the behavior policy based on the offline dataset. Chen et al. (2020) propose a method named Best-Action Imitation Learning (BAIL), which fits a value function, then uses it to select the best actions. Meanwhile, Siegel et al. (2020) propose a method that learns an Advantage-weighted Behavior Model (ABM) and uses it as a prior in performing Maximum a-posteriori Policy Optimization (MPO) (Abdolmaleki et al., 2018). It consists of multiple iterations of policy evaluation and prior learning until they finally perform a policy improvement step using their learned prior to extracting the best possible policy.

Model-based algorithms rely on the offline dataset to learn a dynamics model or a trajectory distribution used for planning. The trajectory distribution induced by models is used to determine the best set of actions to take at each given time step. Kidambi et al. (2020) propose a method named Model-based Offline Reinforcement Learning (MOReL), which measures their model's epistemic uncertainty through an ensemble of dynamics models. Meanwhile, Yu et al. (2020) propose another method named Model-based Offline Policy Optimization (MOPO), which uses the maximum prediction uncertainty from an ensemble of models. Concurrently, Matsushima et al. (2020) propose the BehaviorREgularized Model-ENsemble (BREMEN) method, which learns an ensemble of models of the behavior MDP, as opposed to a pessimistic MDP. In addition, it implicitly constrains the policy to be close to the behavior policy through trust-region policy updates. More recently, Yu et al. (2021a) proposed a method named Conservative Offline Model-Based policy Optimization (COMBO), a model-based version of CQL. The main advantage of COMBO concerning MOReL and MOPO is that it removes the need for uncertainty quantification in model-based offline RL approaches, which is challenging and often unreliable. However, these above offline RL algorithms cannot be directly applied to the offline EF paradigm, which we have described in Section 2 and experimental results empirically verify this claim.

## C  DATASETS

### C.1  DATASET FORMAT

To better introduce the format of our offline EF dataset, we provide an example to show the composition of the offline dataset. According to the data format introduced in the main paper, the data point would be $(s_t, a_t, s_{t+1}, u_{t+1}, d_{t+1})$ and $s_t = (I_1^t, I_2^t, ..., I_n^t, GI, p^t, A(I_{p^t}^t))$. Especially, if the $s_{t+1}$ is the terminal state, i.e., $d_{t+1} = 1$, then we define the $p^{t+1} = -1$ to identify that there is no player need to decide this state. Fig. 12 shows one two-player imperfect-information extensive-form game $\mathcal{G}$. $I_1$ and $I_2$ are information set for Player 1 and Player 2, respectively. If an offline dataset $\mathcal{D}$ covers all state-action pairs, then $\mathcal{D}$ would include the following data points:

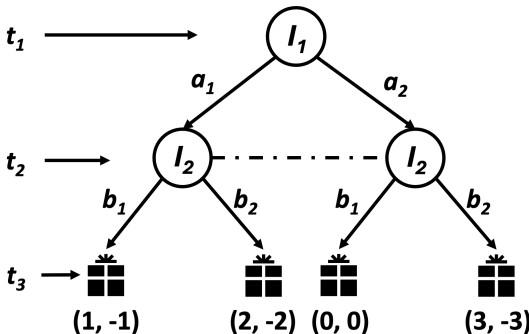

Figure 12: An example game.

$$((I_1^{t_1} = I_1, I_2^{t_1} = \emptyset, GI = \emptyset, 1, \{a_1, a_2\}), a_1, (I_1^{t_2} = I_1 a_1, I_2^{t_2} = I_2, GI = \emptyset, 2, \{b_1, b_2\}), (0, 0), 0),$$

$$((I_1^{t_1} = I_1, I_2^{t_1} = \emptyset, GI = \emptyset, 1, \{a_1, a_2\}), a_2, (I_1^{t_2} = I_1 a_2, I_2^{t_2} = I_2, GI = \emptyset, 2, \{b_1, b_2\}), (0, 0), 0),$$

$$((I_1^{t_2} = I_1 a_1, I_2^{t_2} = I_2, GI = \emptyset, 2, \{b_1, b_2\}), b_1, (I_1^{t_3} = I_1 a_1, I_2^{t_3} = I_2 b_1, GI = \emptyset, -1, \emptyset), (1, -1), 1),$$

$$((I_1^{t_2} = I_1 a_1, I_2^{t_2} = I_2, GI = \emptyset, 2, \{b_1, b_2\}), b_2, (I_1^{t_3} = I_1 a_1, I_2^{t_3} = I_2 b_2, GI = \emptyset, -1, \emptyset), (2, -2), 1),$$

$$((I_1^{t_2} = I_1 a_2, I_2^{t_2} = I_2, GI = \emptyset, 2, \{b_1, b_2\}), b_1, (I_1^{t_3} = I_1 a_2, I_2^{t_3} = I_2 b_1, GI = \emptyset, -1, \emptyset), (0, 0), 1),$$

$$((I_1^{t_2} = I_1 a_2, I_2^{t_2} = I_2, GI = \emptyset, 2, \{b_1, b_2\}), b_2, (I_1^{t_3} = I_1 a_2, I_2^{t_3} = I_2 b_2, GI = \emptyset, -1, \emptyset), (3, -3), 1).$$

We can find that in states $(I_1^{t_2} = I_1 a_1, I_2^{t_2} = I_2, GI = \emptyset, 2, \{b_1, b_2\})$ and $(I_1^{t_2} = I_1 a_2, I_2^{t_2} = I_2, GI = \emptyset, 2, \{b_1, b_2\})$, the information set for Player 2 is the same, as shown in the game tree. Since our dataset is collected from the perspective of the game, we can still distinguish them through the game information of other players and the game information $GI$. Note that there is no chance node in this game, the game information $GI$ is an empty set here. If there is a chance node in the game, the results of the chance node would be recorded into game information $GI$ within the game state and we can distinguish these game states through game information $GI$.

### C.2  VISUALIZATION

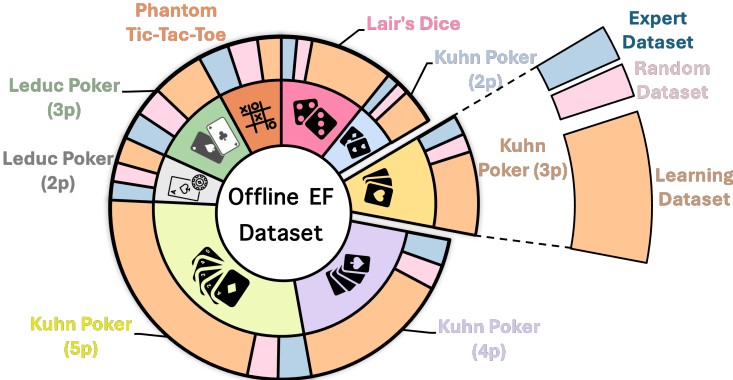

Figure 13: Visualization of the offline EF dataset

Fig. 13 shows a full view of our offline EF dataset. In our offline EF dataset, we collected data for eight games, including two-player Kuhn poker, three-player Kuhn poker, four-player Kuhn poker,

five-player Kuhn poker, two-player Leduc poker, three-player Leduc poker, Phantom Tic-Tac-Toe, and Lair's Dice games. For each game, we generated three datasets, a random dataset, an expert dataset, and a learning dataset, following our data collection methods. To validate the diversity of these collected offline datasets and gain insights into them, we also introduce a visualization method for comparing them. Firstly, we generate the game tree for the corresponding game. Subsequently, we traverse the game tree using depth-first search (DFS) (Tarjan, 1972) and assign an index to each leaf node based on the DFS results. Then, we count the frequency of each leaf node within the dataset. The reason why we do this is that each leaf node represents a unique sampled trajectory originating from the root node of the game tree. As a result, the frequency of leaf nodes can effectively capture the distribution of the dataset. Finally, these frequency data can be plotted to visualize. Fig. 14 visualizes some datasets of some games. From these figures, we can find that in the random dataset, the frequency of leaf nodes is nearly uniform, whereas, in the expert dataset, the frequency of leaf nodes is uneven. The distribution of the learning dataset and the hybrid dataset falls between that of the expert dataset and the random dataset. These observations confirm that the distribution of these datasets differs, thus validating the diversity of our proposed offline datasets.

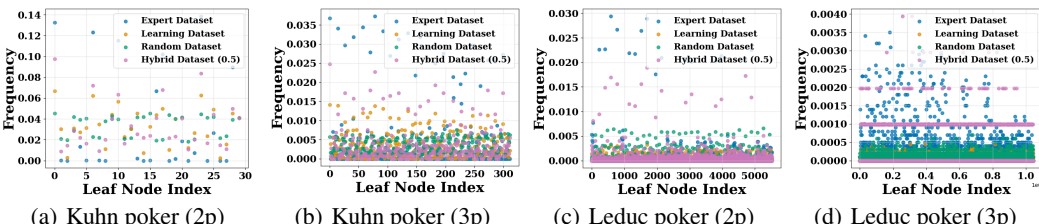

    (a) Kuhn poker (2p)    (b) Kuhn poker (3p)    (c) Leduc poker (2p)    (d) Leduc poker (3p)

Figure 14: Frequency of leaf node in different offline datasets.

## D    Theoretical Analysis

In this section, we provide a comprehensive theoretical analysis of the offline EF paradigm and our BOMB framework to facilitate the understanding of the offline EF paradigm and BOMB framework. We first provide the minimal dataset assumption that is sufficient to compute the equilibrium strategy in the offline setting. Then we provide a general generalization bound for training neural network models. Finally, we give the performance guarantee for our algorithm. In the following sections, we assume that all extensive-form games discussed here are perfect recall and timetable.

### D.1    Minimal Dataset Assumption for offline EF

As demonstrated in offline RL papers (Rashidinejad et al., 2021; Xie et al., 2021), a dataset coverage condition over the optimal policy is sufficient for offline learning. Therefore, it is straightforward to extend this dataset coverage assumption to the offline EF paradigm. In the main paper, we have proved that the dataset generated by the equilibrium strategy is not sufficient for computing the equilibrium strategy in an offline manner by providing a counter-example. Furthermore, we also provide another dataset assumption related to the equilibrium strategy, shown in the following assumption.

**Assumption D.1.** (Single Strategy Coverage) The offline dataset $\mathcal{D}$ is said to be *single strategy coverage* if the equilibrium strategy profile $\sigma^*$ is covered by the offline dataset $\mathcal{D}$, i.e., for each player $i$, each information set $I_i$, and action $a_i$ with $\sigma_i^*(I_i, a_i) > 0$, there is a corresponding state-action pair $(s_t, a_i)$ in $D$.

Subsequently, a question arises: *is the single strategy coverage assumption also sufficient for computing equilibrium strategy in the offline setting?* We employ the following theorem to answer this question and elucidate the rationale behind this.

**Theorem D.2.** *Single strategy coverage assumption over offline dataset $\mathcal{D}$ is not sufficient for computing computing an $\epsilon$-equilibrium for an arbitrarily small $\epsilon$ in the offline setting.*

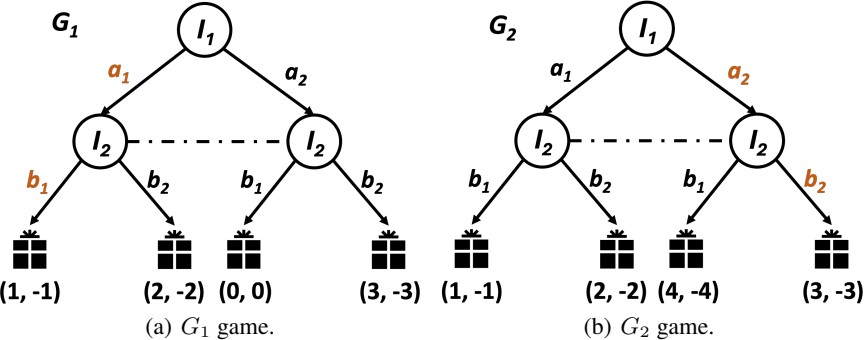

(a) $G_1$ game.        (b) $G_2$ game.

Figure 15: Counter-example for proving Theorem D.2.

*Proof.* We prove this theorem by providing a counter-example. To this end, we consider two two-player IIEFGs $G_1$ and $G_2$, represented in Fig. 15. We can easily find that the NE of the game $G_1$ is strategy profile $\sigma^1 = (\sigma_1^1, \sigma_2^1) = (\{I_1 : a_1\}, \{I_2 : b_1\})$, i.e., Player 1 plays $a_1$ at information set $I_1$ and Player 2 plays $b_1$ at information set $I_2$. The NE of the game $G_2$ is strategy profile $\sigma^2 = (\sigma_1^2, \sigma_2^2) = (\{I_1 : a_2\}, \{I_2 : b_2\})$. Now we consider an offline dataset $\mathcal{D}$ which is generated using a strategy profile $\sigma_{\mathcal{D}}$. The $\sigma_{\mathcal{D}}$ is set to be the uniform distribution over the strategy profiles $\sigma^1$ and $\sigma^2$, which means that dataset $\mathcal{D}$ covers both $\sigma^1$ and $\sigma^2$. Therefore, the offline dataset $\mathcal{D}$ satisfies the single strategy coverage assumption for these two games $G_1$ and $G_2$. However, no algorithm can distinguish these two extensive-form games only based on dataset $\mathcal{D}$ since these two games are both consistent on dataset $\mathcal{D}$. In conclusion, the single strategy converges assumption is not sufficient for computing an $\epsilon$-equilibrium for an arbitrarily small $\epsilon$ in the offline setting.    □

From the above proof, we know that the single strategy coverage assumption is sufficient for computing the optimal strategy in the offline RL setting while it is not sufficient for computing an NE

strategy in the offline setting. The intuition behind this is that in an offline RL setting, we can easily use the data of two actions to decide which action is better, whereas, in the offline EF paradigm, we cannot use data from only two action pairs to know which action pair is closer to NE, because identifying NE requires other action pairs as inferences. Based on this analysis, Cui & Du (2022) provide a minimal coverage assumption which is sufficient for computing an NE strategy in the two-player zero-sum Markov games, which is defined as follows,

**Assumption D.3.** (Deterministic Unilateral Coverage) For all deterministic strategy $\sigma_i$ for player $i$, $(\sigma_i, \sigma^*_{-i})$ are covered by the dataset, where $(\sigma^*_1, ..., \sigma^*_n)$ is one NE strategy.

**Assumption D.4.** (Unilateral Coverage) For all (possible stochastic) strategy $\sigma_i$ for all player $i$, $(\sigma_i, \sigma^*_{-i})$ are covered by the dataset, where $(\sigma^*_1, ..., \sigma^*_n)$ is one NE strategy.

Note that deterministic unilateral coverage assumption is equivalent to unilateral coverage assumption. The intuition behind this is that any mixed strategy can be represented by a combination of deterministic strategies. Therefore, if all deterministic strategies are covered by the dataset, then all mixed strategies are also covered. Based on this finding, in the following proof, we only consider all deterministic strategies. Previously, Cui & Du (2022) established that unilateral coverage assumption is the minimal sufficient condition for computing an NE strategy in the two-player zero-sum Markov games. However, this unilateral coverage assumption over the offline dataset is **not sufficient** for our model-based method to compute the equilibrium strategy in the offline setting. We formally proved this limitation through the following theorem.

**Theorem D.5.** *The unilateral coverage assumption over offline dataset $\mathcal{D}$ is not sufficient for our model-based method to converge to an $\epsilon$-equilibrium for an arbitrarily small $\epsilon$ in the offline setting.*

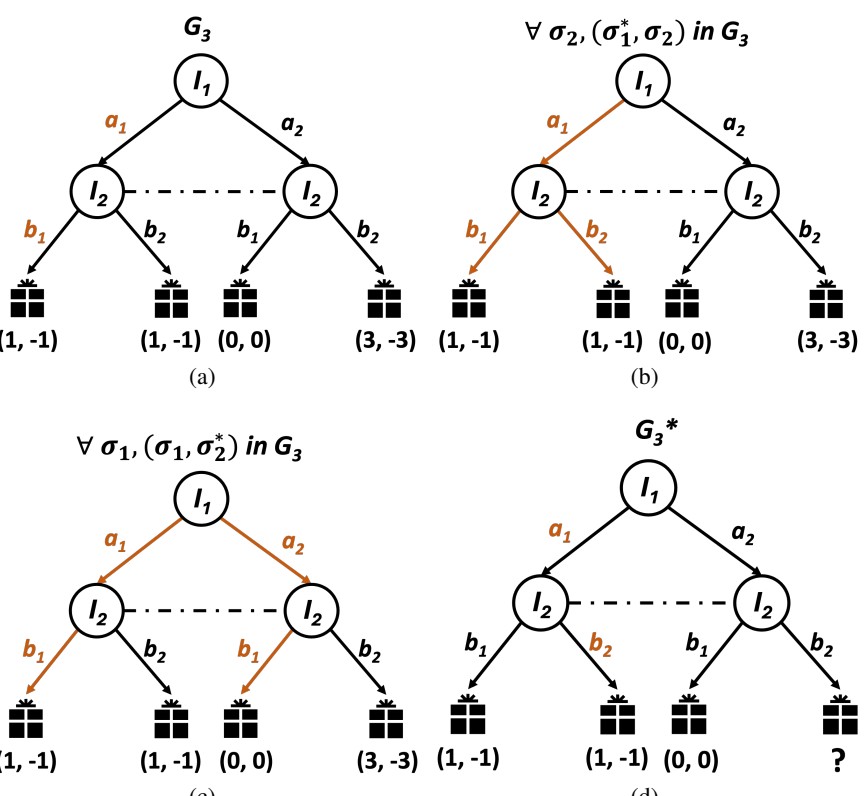

Figure 16: Counter-example for proving Theorem D.5.

*Proof.* We prove this theorem by providing a counter-example. First, we consider an IIEFG $M_3$, represented in Fig. 16(a). We can easily find that the NE strategy of game $G_3$ is strategy profile $\sigma^* = (\sigma_1, \sigma_2) = (\{I_1 : a_1\}, \{I_2 : b_1\})$. To build a dataset $\mathcal{D}$ satisfying the unilateral coverage assumption, the dataset needs to cover $(\sigma^*_1, \sigma_2)$ for all deterministic strategy $\sigma_2$ and $(\sigma_1, \sigma^*_2)$ for

all deterministic strategy $\sigma_1$. We show the state-action pairs covered by these strategy profiles in Figs. 16(b)-16(c). It means that if the dataset $\mathcal{D}$ satisfies the unilateral coverage assumption, then the dataset $\mathcal{D}$ would cover these state-action pairs marked by these orange lines. When applying our model-based method on the dataset $\mathcal{D}$, the first step is to train an environment model based on the dataset $\mathcal{D}$. Assume that the environment model can be trained well (i.e., Assumption 4.3 holds) which means that the environment model can precisely represent all game information in the dataset. Therefore, the game represented by the trained environment model would be $G_3^*$ in Fig. 16(d). Note that there is some missing data in the game. Although our trained environment model can give approximate results for these missing data, it may result in a different equilibrium strategy. For example, if the missing value in $G_3^*$ is $(0,0)$ or $(-1,1)$, then the strategy profile $\sigma = (\sigma_1, \sigma_2') = (\{I_1 : a_1\}, \{I_2 : b_2\})$ would be the NE strategy of game $G_3^*$. However, the strategy profile $\sigma$ is not the NE strategy for the original game $G_3$. Therefore, the unilateral coverage assumption over the dataset is not sufficient for our model-based method to converge to to an $\epsilon$-equilibrium for an arbitrarily small $\epsilon$. □

To guarantee the convergence of our model-based method, we provide a minimal dataset coverage assumption for our model-based method to converge to the equilibrium strategy of the original game under the offline setting.

**Definition D.6** (Definition 4.1). An offline dataset $\mathcal{D}$ is said to be a *uniform coverage* of an IIEFG $\mathcal{G}$ if and only if the offline dataset $\mathcal{D}$ covers all possible state-action pairs. Formally, $(s_t, a_t, s_{t+1}, u_{t+1}, d_{t+1}), \forall s_t, a_t \in A(s_t)$ and $s_{t+1} \in T(s_t, a_t)$ where $T$ is the transition function of game $\mathcal{G}$.

**Theorem D.7** (Theorem 4.4). *Let $\sigma_{MB(\mathcal{D})}$ be the strategy profile learned by our **model-based algorithm** based on the offline dataset $\mathcal{D}$ with sufficient data under Assumption 4.3. Then, $\sigma_{MB(\mathcal{D})}$ is guaranteed to be an $\epsilon$-equilibrium strategy of the IIEFG $\mathcal{G}$ if and only if $\mathcal{D}$ is a uniform coverage of $\mathcal{G}$ and $\sigma_{MB(\mathcal{D})}$ is an $\epsilon$-equilibrium strategy for the trained environment model within the model-based algorithm.*

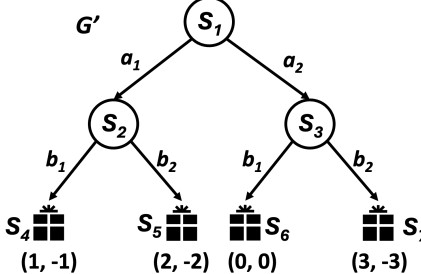

Figure 17: $G'$ Game.

*Proof.* From the example in the proof of Theorem D.5, we find that a slight violation of the uniform coverage assumption, i.e., only one state-action pair is missing, will impede the computation of the equilibrium strategy using our model-based method. In other words, any state-action pair that is not covered by the dataset may cause failure in computing the equilibrium strategy of the original game using our model-based method.

Next, we need to prove that the dataset satisfying the uniform coverage assumption can guarantee the convergence to the equilibrium strategy of the original game using our model-based method. In our model-based method, we need to train an environment model based on the offline dataset. Therefore, to prove the convergence guarantee under the uniform coverage dataset assumption, we need to verify whether the game reconstructed from the dataset satisfying the uniform coverage assumption is the same as the original game. Here, we reuse the example in the App. C.1. In that example, the offline datset $\mathcal{D}$ of the IIEFG $\mathcal{G}$ covers all state-action pairs. Therefore, the offline dataset $\mathcal{D}$ satisfies the uniform coverage dataset assumption. From the offline dataset $\mathcal{D}$, we can

easily rebuild the game $G'$, as shown in Fig. 17. In the game $G'$,

$$S_1 = (I_1^{t_1} = I_1, I_2^{t_1} = \emptyset, GI = \emptyset, 1, \{a_1, a_2\}), S_2 = (I_1^{t_2} = I_1 a_1, I_2^{t_2} = I_2, GI = \emptyset, 2, \{b_1, b_2\}),$$

$$S_3 = (I_1^{t_2} = I_1 a_2, I_2^{t_2} = I_2, GI = \emptyset, 2, \{b_1, b_2\}), S_4 = (I_1^{t_3} = I_1 a_1, I_2^{t_3} = I_2 b_1, GI = \emptyset, -1, \emptyset),$$

$$S_5 = (I_1^{t_3} = I_1 a_1, I_2^{t_3} = I_2 b_2, GI = \emptyset, -1, \emptyset), S_6 = (I_1^{t_3} = I_1 a_2, I_2^{t_3} = I_2 b_1, GI = \emptyset, -1, \emptyset),$$

$$S_7 = (I_1^{t_3} = I_1 a_2, I_2^{t_3} = I_2 b_2, GI = \emptyset, -1, \emptyset).$$

Especially, for game states $S_2$ and $S_3$, the player acting is both Player 2 and the information set for Player 2 is the same. Therefore, these two game states correspond to different game nodes under the same information set. Although Player 2 cannot distinguish these two game states, from the perspective of the game, we can still distinguish them by the information set of Player 1. Particularly, if there is a chance node in the game, the result of the chance node would be recorded in $GI$ within the game state $S$. Therefore, we can still distinguish these game states by game information $GI$. Since the dataset satisfying the uniform coverage assumption covers all state-action pairs, the links between game states can be built following these data points in the dataset. According to Assumption 4.3, the error in training the environment game model based on $\mathcal{D}$ can be considered negligible. Consequently, the trained environment game model is identical to the original game $\mathcal{G}$, as the dataset $\mathcal{D}$ provides full coverage of all state transitions. Therefore, we can find that the reconstructed game tree has the same game states and the same transition function as the original game, thereby the same equilibrium strategy. Therefore, our reconstructed game model can provide the same information as the underlying game of the offline dataset. Then applying our model-based equilibrium finding algorithm to the reconstructed game model definitely can converge to the equilibrium strategy of the underlying game in the offline setting. Formally, if $\sigma_{MB(\mathcal{D})}$ is an $\epsilon$-equilibrium strategy for the trained environment game model, it is also an $\epsilon$-equilibrium strategy for the original game $\mathcal{G}$. □

So far, we have proved that the uniform dataset coverage assumption is sufficient for our model-based method to converge to the equilibrium strategy under the offline setting. For our behavior cloning method, these dataset coverage assumptions may not be sufficient to converge to the equilibrium strategy since its performance mainly depends on the underlying behavior strategy of the dataset. In the following theorem, we provide a minimal dataset coverage assumption for our behavior cloning method to converge to the equilibrium strategy in the offline setting.

**Definition D.8** (Definition 4.2). An offline dataset $\mathcal{D}$ is said to be an $\epsilon$-*equilibrium coverage* over an IIEFG $\mathcal{G}$ if and only if its underlying behavior strategy $\sigma_{\mathcal{D}}$ satisfies $\text{GAP}(\sigma_{\mathcal{D}}, \sigma^*) < \epsilon$, where $\sigma_{\mathcal{D}}$ is defined as $\sigma_{\mathcal{D}}(s_t, a_t) = \frac{C(s_t, a_t)}{C(s_t)}$ and $\sigma_D(s_t, a_t) > 0$ for all $s_t$ and $a_t \in A(s_t)$, with $C(s_t, a_t)$ and $C(s_t)$ denoting the counts of data points containing $(s_t, a_t)$ and $s_t$ in $\mathcal{D}$, respectively.

This definition ensures that the unique correspondence relationship between the equilibrium-covered dataset and the equilibrium strategy. Specifically, the dataset is generated by the equilibrium strategy and the strategy represented by the dataset would be the same as the equilibrium strategy.

**Theorem D.9** (Theorem 4.5). *Let $\sigma_{BC(\mathcal{D})}$ be the strategy profile learned by our **behavior cloning algorithm** based on the offline dataset $\mathcal{D}$ with sufficient data under Assumption 4.3. Then $\sigma_{BC(\mathcal{D})}$ is guaranteed to be an $\epsilon$-equilibrium strategy of IIEFG $\mathcal{G}$ if and only if the offline dataset $\mathcal{D}$ is an $\epsilon$-equilibrium coverage of the IIEFG $\mathcal{G}$.*

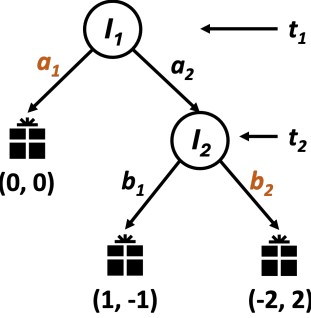

Figure 18: Game example.

*Proof.* According to Assumption 4.3, the error in training the behavior cloning strategy $\sigma_{BC(\mathcal{D})}$ from the dataset $\mathcal{D}$ is negligible. Therefore, by the behavior cloning process, $\sigma_{BC(\mathcal{D})}$ is identical to the behavior strategy underlying $\mathcal{D}$, i.e., $\sigma_{BC(\mathcal{D})} = \sigma_{\mathcal{D}}$. Consequently, if $\mathcal{D}$ is an $\epsilon$-equilibrium coverage of $\mathcal{G}$, then $\sigma_{BC(\mathcal{D})}$ is an $\epsilon$-equilibrium strategy for the IIEFG $\mathcal{G}$, and vice visa, as $\text{GAP}(\sigma_{\mathcal{D}}, \sigma^*) < \epsilon$ if and only if $\text{GAP}(\sigma_{BC(\mathcal{D})}, \sigma^*) < \epsilon$. Next, we prove that any slight violation of these conditions would invalidate the convergence result.

Here, we reuse the example in Section 4.2, as shown in Fig. 18. Note that the NE strategy of the game is a pure strategy, i.e, $\sigma^* = (\sigma_1^*, \sigma_2^*) = (\{I_1 : a_1\}, \{I_2 : b_2\})$. If we use this equilibrium strategy to generate the offline dataset $\mathcal{D}$, then $\mathcal{D}$ would only include the data point $((I_1^{t_1} = I_1, I_2^{t_1} = \emptyset, GI = \emptyset, 1, \{a_1, a_2\}), a_1, (I_1^{t_2} = I_1 a_1, I_2^{t_2} = \emptyset, GI = \emptyset, -1, \emptyset), (0, 0), 1)$. We cannot get the equilibrium strategy only from $\mathcal{D}$. In this example game, the offline dataset $\mathcal{D}$ is generated by a pure equilibrium strategy instead of a fully mixed equilibrium strategy, and the behavior cloning method cannot get the equilibrium strategy from the offline dataset $\mathcal{D}$ since there is no information about Player 2. Another example is the dataset $D'$ covering the equilibrium strategy $\sigma^*$, i.e., the $D'$ includes the data points

$$((I_1^{t_1} = I_1, I_2^{t_1} = \emptyset, GI = \emptyset, 1, \{a_1, a_2\}), a_1, (I_1^{t_2} = I_1 a_1, I_2^{t_2} = \emptyset, GI = \emptyset, -1, \emptyset), (0, 0), 1),$$

$$((I_1^{t_1} = I_1, I_2^{t_1} = \emptyset, GI = \emptyset, 1, \{a_1, a_2\}), a_2, (I_1^{t_2} = I_1 a_2, I_2^{t_2} = I_2, GI = \emptyset, 2, \{b_1, b_2\}), (0, 0), 0),$$

$$((I_1^{t_2} = I_1 a_2, I_2^{t_2} = I_2, GI = \emptyset, 2, \{b_1, b_2\}), b_2, (I_1^{t_2} = I_1 a_2, I_2^{t_2} = I_2 b_2, GI = \emptyset, -1, \emptyset), (-2, 2), 1).$$

To cover the equilibrium strategy of Player 2, the data point $((I_1^{t_1} = I_1, I_2^{t_1} = \emptyset, GI = \emptyset, 1, \{a_1, a_2\}), a_2, (I_1^{t_2} = I_1 a_2, I_2^{t_2} = I_2, GI = \emptyset, 2, \{b_1, b_2\}), (0, 0), 0)$ should also be visited. Although $D'$ covers the equilibrium strategy, $D'$ does not satisfy the $\epsilon$-equilibrium coverage assumption since the $D'$ dose not created by the $\epsilon$-equilibrium strategy. Then the behavior cloning method cannot converge to the equilibrium strategy $\sigma^*$ based on $D'$ since BC cannot get the pure strategy for Player 1 under the influence of the data point $((I_1^{t_1} = I_1, I_2^{t_1} = \emptyset, GI = \emptyset, 1, \{a_1, a_2\}), a_2, (I_1^{t_2} = I_1 a_2, I_2^{t_2} = I_2, GI = \emptyset, 2, \{b_1, b_2\}), (0, 0), 0)$. Therefore, a slight violation of the equilibrium coverage assumption would cause failure in computing the $\epsilon$-equilibrium strategy of the original game using our behavior cloning method. In conclusion, the equilibrium coverage assumption is the minimal dataset coverage assumption that guarantees the convergence to the equilibrium strategy of the original game using our behavior cloning method. Formally, $\sigma_{BC(\mathcal{D})}$ is guaranteed to be an $\epsilon$-equilibrium strategy of IIEFG $\mathcal{G}$ if and only if the offline dataset $\mathcal{D}$ is an $\epsilon$-equilibrium coverage of the IIEFG $\mathcal{G}$. $\qquad\square$

**Theorem D.10** (Theorem 4.6). *Let $\sigma_{BOMB(\mathcal{D})}$ represent the strategy profile learned by our **BOMB** algorithm based on the offline dataset $\mathcal{D}$ with sufficient data under Assumption 4.3, $\sigma_{\mathcal{D}}$ represent the underlying behavior strategy of $\mathcal{D}$ and $\sigma^*$ represent the equilibrium strategy of IIEFG $\mathcal{G}$. Then the gap between $\sigma_{BOMB(\mathcal{D})}$ and $\sigma^*$ is at most equal to, or smaller than, the gap between $\sigma_{\mathcal{D}}$ and $\sigma^*$, i.e., $\text{GAP}(\sigma_{BOMB(\mathcal{D})}, \sigma^*) \leq \text{GAP}(\sigma_{\mathcal{D}}, \sigma^*)$.*

*Proof.* According to Assumption 4.3, the error in training the behavior cloning strategy $\sigma_{BC(\mathcal{D})}$ from the dataset $\mathcal{D}$ is negligible. Therefore, by the behavior cloning process, $\sigma_{BC(\mathcal{D})}$ is identical to the behavior strategy underlying $\mathcal{D}$, i.e., $\sigma_{BC(\mathcal{D})} = \sigma_{\mathcal{D}}$. Then $\text{GAP}(\sigma_{BOMB(\mathcal{D})}, \sigma^*) = \text{GAP}(\sigma_{\mathcal{D}}, \sigma^*)$ if $\alpha = 1$ in our **BOMB algorithm**.

If the dataset satisfies the uniform coverage, by Theorem 4.4, $\text{GAP}(\sigma_{BOMB(\mathcal{D})}, \sigma^*) \leq \text{GAP}(\sigma_{\mathcal{D}}, \sigma^*)$ if $\alpha = 0$ in our **BOMB algorithm**.

Therefore, in general case, $\text{GAP}(\sigma_{BOMB(\mathcal{D})}, \sigma^*) \leq \text{GAP}(\sigma_{\mathcal{D}}, \sigma^*)$. $\qquad\square$

## D.2 GENERALIZATION BOUND FOR TRAINING MODEL

As described in the main paper, to conduct the BOMB framework, we need to train one behavior cloning policy and an environment model which are both neural network models. Furthermore, these two models are trained in a supervised learning manner with different loss functions based on the offline EF dataset. Here, we provide a general generalization bound for training such neural network models facilitating the following analysis of the BOMB framework.

As we know, the supervised learning framework includes a data-generation distribution $\sigma$, a hypothesis class $\mathcal{H}$ of the neural network approximator, a training dataset $\mathcal{D}$, and evaluation metrics to evaluate the performance of any approximator. Here, we can use the loss function $l$ to evaluate the performance of any approximation. The learning framework aims to minimize the true risk function $L_\sigma(h)$ which is the expected loss function of $h \in \mathcal{H}$ under the distribution $\sigma$,

$$L_\sigma(h) = \mathbb{E}_{d \sim \sigma}[l(h(d), d)].$$

Accordingly, the empirical risk function $L_\mathcal{D}(h)$ on the training dataset $\mathcal{D}$ can be defined as:

$$L_\mathcal{D} = \frac{1}{|\mathcal{D}|} \sum_{d \sim \mathcal{D}} [l(h(d), d)].$$

To get a generalization bound, we use an auxiliary lemma from (Shalev-Shwartz & Ben-David, 2014). Therefore, we can measure the capacity of the composition function class $l \circ \mathcal{H}$ using the empirical Rademacher complexity on the training set $\mathcal{D}$ with size $m$, which is defined as:

$$\mathcal{R}_\mathcal{D}(l \circ \mathcal{H}) = \frac{1}{m} \mathbb{E}_{\mathbf{x} \sim \{+1, -1\}^m} [\sup_{h \in \mathcal{H}} \sum_{i=1}^{m} x_i \cdot l(h(d_i), d_i)]$$

where $\mathbf{x}$ is distributed i.i.d. according to uniform distribution in $\{+1, -1\}$. Before providing the generalization bound, we first provide the distance between two different approximators and one common theorem to facilitate the proof of the generalization bound.

**Definition D.11.** ($r$-cover) We say function class $\mathcal{H}_r$ $r$-cover $\mathcal{H}$ under $\ell_{\infty,1}$-distance if $\forall h, h \in \mathcal{H}$, there exists $h_r$ in $\mathcal{H}_r$ such that $||h - h_r||_{\infty,1} = \max_{x \in \mathcal{D}} ||h(x) - h_r(x)||_1 \le r$.

**Definition D.12.** ($r$-covering number) The $r$-covering number of $\mathcal{H}$, $\mathcal{N}_{\infty,1}(\mathcal{H}, r)$, is the cardinality of the smallest function class $H_r$ that $r$-covers $\mathcal{H}$ under $\ell_{\infty,1}$-distance.

**Theorem D.13.** *(Shalev-Shwartz & Ben-David, 2014) Let $\mathcal{D}$ be a training set of size $m$ drawn i.i.d. from distribution $\sigma$. Then with probability of at least $1 - \delta$ over draw of $\mathcal{D}$ from $\sigma$, for all $h \in \mathcal{H}$,*

$$L_\sigma(h) - L_\mathcal{D}(h) \le 2\mathcal{R}_\mathcal{D}(l \circ \mathcal{H}) + 4\sqrt{\frac{2\ln(4/\delta)}{m}}.$$

We provide the bound to measure the generalizability of the trained approximator in a training dataset with size $m$.

**Theorem D.14** (Generalization bound). *Assume that the loss function $l$ is $T$-Lipschitz continuous, then for hypothesis class $\mathcal{H}$ of approximator and distribution $\sigma$, with probability at least $1 - \delta$ over draw of the training set $\mathcal{D}$ with size $m$ from $\sigma$, for all $h \in \mathcal{H}$, we have*

$$L_\sigma(h) - L_\mathcal{D}(h) \le 2 \cdot \inf_{r > 0} [\frac{\sqrt{2\log \mathcal{N}_{\infty,1}(\mathcal{H}, r)}}{m} + Tr] + 4\sqrt{\frac{2\ln(4/\delta)}{m}}.$$

*Proof.* According to Theorem D.13, we have

$$L_\sigma(h) - L_\mathcal{D}(h) \le 2\mathcal{R}_\mathcal{D}(l \circ \mathcal{H}) + 4\sqrt{\frac{2\ln(4/\delta)}{m}}.$$

According to the assumption, the loss function $l(x, y)$ is $T$-Lipschitz continuous under $\ell_k$-distance, i.e., $|l(x, y) - l(x', y)| \le T||x - x'||_k$, where $|| \cdot ||_k$ is the $k$-norm. Let $\mathcal{H}_r$ be the function class that $r$-cover $\mathcal{H}$ for some $r > 0$ and $|\mathcal{H}_r| = \mathcal{N}_{\infty,1}(\mathcal{H}, r)$ be the $r$-covering number of $\mathcal{H}_r$. For all $h \in \mathcal{H}$, $h_r \in \mathcal{H}_r$ is denoted to be the function approximator that $r$-covers $h$. Based on above equation, we have

$$|l(h(x), y) - l(h_r(x), y)| \le T||h(x) - h_r(x)||_k \le Tr.$$

Then we have

$$\mathcal{R}_\mathcal{D}(l \circ \mathcal{H}) = \frac{1}{m} \mathbb{E}_{\mathbf{x} \sim \{+1,-1\}^m} [\sup_{h \in \mathcal{H}} \sum_{i=1}^m x_i \cdot l(h(d_i), d_i)] \tag{1}$$

$$= \frac{1}{m} \mathbb{E}_{\mathbf{x} \sim \{+1,-1\}^m} [\sup_{h \in \mathcal{H}} \sum_{i=1}^m x_i \cdot (l(h_r(d_i), d_i) + l(h(d_i), d_i) - l(h_r(d_i), d_i))] \tag{2}$$

$$\leq \frac{1}{m} \mathbb{E}_{\mathbf{x} \sim \{+1,-1\}^m} [\sup_{h_r \in \mathcal{H}_r} \sum_{i=1}^m x_i \cdot l(h_r(d_i), d_i)] + \frac{1}{m} \mathbb{E}_{\mathbf{x} \sim \{+1,-1\}^m} [\sup_{h \in \mathcal{H}} \sum_{i=1}^m |x_i \cdot Tr|] \tag{3}$$

$$\leq \sup_{h_r \in \mathcal{H}_r} \sqrt{\sum_{i=1}^m (\ell(h_r, d_i))^2} \cdot \frac{\sqrt{2 \log \mathcal{N}_{\infty,1}(\mathcal{H}, r)}}{m} + \frac{Tr}{m} \mathbb{E}_{\mathbf{x}} ||\mathbf{x}||_1 \tag{4}$$

$$\leq \frac{\sqrt{2 \log \mathcal{N}_{\infty,1}(\mathcal{H}, r)}}{m} + Tr \tag{5}$$

The reduction from Eq. 3 to Eq. 4 is based on Massart's lemma (Shalev-Shwartz & Ben-David, 2014). Finally,

$$L_\sigma(h) - L_\mathbf{D}(h) \leq 2\mathcal{R}_\mathcal{D}(l \circ \mathcal{H}) + 4\sqrt{\frac{2 \ln (4/\delta)}{m}} \leq 2 \cdot \inf_{r>0} [\frac{\sqrt{2 \log \mathcal{N}_{\infty,1}(\mathcal{H}, r)}}{m} + Tr] + 4\sqrt{\frac{2 \ln (4/\delta)}{m}}$$

$$\square$$

Therefore, given a training dataset with size $m$, we can have a generalization bound for the error depending on the characteristic of the loss function. In this paper, we follow the supervised learning framework to train the behavior cloning policy and environment model. Therefore, we can provide the following assumptions for the trained policy and environment models based on the above theorem.

**Assumption D.15.** Suppose the error for training the behavior cloning policy is less than an extremely small $\epsilon$ on the dataset with enough data (the size of data can be computed according to the above theorem). In that case, we consider that the trained behavior cloning policy is the same as the underlying behavior strategy of the dataset.

**Assumption D.16.** Suppose the error for training the environment model is less than an extremely small $\epsilon$ on the dataset with enough data. In that case, we consider that the trained environment model can provide the full information for the underlying game of the dataset.

# E    IMPLEMENTATION DETAILS

Here, we provide the details for the model-based method by introducing our instantiate algorithms: MB-PSRO and MB-CFR, which are adaptions from two widely-used online equilibrium finding algorithms PSRO and Deep CFR.

## E.1    MB-PSRO

---

**Algorithm 2** MB-PSRO

---

1: **Input:** Trained environment model $E_{\theta_e}$
2: Initial policy sets $\Pi$ for all players;
3: Compute expected rewards $U^\Pi$ for each strategy $\pi \in \Pi$ *based on the environment model $E_{\theta_e}$*;
4: Initialize mate-strategies $\sigma_i = \text{UNIFORM}(\Pi_i)$, $\forall i$;
5: **repeat**
6:     **for** each player $i \in [1, .., n]$ **do**
7:         **for** best response episodes $t \in [1, ..., T]$ **do**
8:             Sample $\pi_{-i} \sim \sigma_{-i}$;
9:             Train best response policy $\pi_i'$ over $\rho \sim (\pi_i', \pi_{-i})$, which *samples on the environment model $E_{\theta_e}$*;
10:        **end for**
11:        add the best response policy $\pi_i'$ to policy set $\Pi_i$;
12:    **end for**
13:    Compute missing entries in $U^\Pi$ *based on the environment model $E_{\theta_e}$*;
14:    Compute the meta-strategy $\sigma$ using any meta-solver;
15: **until** Meet the convergence condition
16: **Output:** Policy set $\Pi$ and meta-strategy $\sigma$

---

We present the whole framework in Alg. 2. In the beginning, we need the well-trained environment model $E_{\theta_e}$ as input to replace the function of the actual environment. Firstly, we initialize policy sets $\Pi$ for all players using random strategies. Then, we estimate the expected utilities for each strategy profile based on the environment model $E_{\theta_e}$ to form the meta-game matrix. In vanilla PSRO, this process needs to interact with the actual game environment. However, in the offline setting, the actual game environment is not available. Therefore, we use the well-trained environment model $E_{\theta_e}$ to replace the actual game environment to provide the information needed in the algorithm. After building the meta-game matrix, the meta-strategy is initialized by a uniform strategy. Next, we compute the best response policy for every player and add these trained best response policies to their policy sets. When training the best response policy oracle using DQN or other RL algorithms, we sample the training data based on the environment model $E_{\theta_e}$. After adding these trained best response policies, we compute missing entries in the meta-game matrix still based on the trained environment model $E_{\theta_e}$. Then, the meta-strategy $\sigma$ of the meta-game matrix can be computed using any meta-solver, such as Nash solver or $\alpha$-rank algorithm. For games with more than two players, the $\alpha$-rank algorithm is taken as the meta-solver. Finally, we repeat the above processes until meeting the convergence condition and output the policy set and meta-strategy as the approximate equilibrium strategy.

To compute the CCE strategy, we also instantiate one algorithm: MB-JPSRO, an adaptation from the JPSRO algorithm. The process of JPSRO is similar to PSRO except for the best response computation and meta solver. Therefore, MB-JPSRO is also similar to MB-PSRO. For this reason, we do not cover MB-JPSRO in detail here.

## E.2    MB-CFR

Alg. 3 shows the process of MB-CFR, which is adapted from the Deep CFR algorithm. It also needs the well-trained environment model $E_{\theta_e}$ as input for the MB-CFR algorithm. We first initialize regret and strategy networks for each player and then initialize regret and strategy memories for each player (Lines 2-4). Then we need to update the regret network for every player. To do this, we perform a traverse function to collect corresponding training data. The traverse function can be any sampling-based CFR algorithm. Here, we use the external sampling algorithm as the traverse

**Algorithm 3** MB-CFR

1: **Input:** Trained environment model $E_{\theta_e}$
2: Initialize regret network $R(I, a|\theta_{r,p})$ for all players;
3: Initialize strategy network $S(I|\theta_{\pi,p})$ for all players;
4: Initialize regret memory $M_{r,p}$ and strategy memory $M_{\pi,p}$ for every player $p$;
5: **for** iteration $t = 1$ to $T$ **do**
6:     **for** player $p \in [1, ..., n]$ **do**
7:         **for** traverse episodes $k \in [1, ..., K]$ **do**
8:             **TRVERSE**($\phi, p, \theta_{r,p}, \theta_{\pi,-p}, M_{r,p}, M_{\pi,-p}, E_{\theta_e}$);
            # Use sample algorithm to traverse the game tree and record regret and strategy training data
9:         **end for**
10:         Train $\theta_{r,p}$ from scratch based on regret memory $M_{r,p}$ for every player $p$;
11:     **end for**
12: **end for**
13: Train $\theta_{\pi,p}$ based on strategy memory $M_{\pi,p}$ for every player $p$;
14: **Output:**$\theta_{\pi,p}$ for every player $p$

---

**Algorithm 4** TRVERSE($s, p, \theta_{r,p}, \theta_{\pi,-p}, M_{r,p}, M_{\pi,-p}, E_{\theta_e}$)-External Sampling Algorithm

1: **if** $s$ is terminal state **then**
2:     Get the utility $u_p(s)$ from the environment model $E_{\theta_e}$;
3:     **Output:** $u_p(s)$
4: **else if** $s$ is a chance state **then**
5:     Sample an action $a$ from the available actions, which is obtained from model $E_{\theta_e}$;
6:     $s' = E_{\theta_e}(s, a)$;
7:     **Output:**TRAVERSE($s', p, \theta_{r,p}, \theta_{\pi,-p}, M_{r,p}, M_{\pi,-p}, E_{\theta_e}$)
8: **else if** $P(s) = p$ **then**
9:     $I \leftarrow s[p]$; # Get the corresponding information set from the game state
10:     $\sigma(I) \leftarrow$ strategy of $I$ computed using regret values $R(I, a|\theta_{r,p})$ based on regret matching;
11:     **for** $a \in A(s)$ **do**
12:         $s' = E_{\theta_e}(s, a)$;
13:         $u(a) \leftarrow$ TRAVERSE($s', p, \theta_{r,p}, \theta_{\pi,-p}, M_{r,p}, M_{\pi,-p}, E_{\theta_e}$);
14:     **end for**
15:     $u_\sigma \leftarrow \sum_{a \in A(s)} \sigma(I, a)u(a)$;
16:     **for** $a \in A(s)$ **do**
17:         $r(I, a) \leftarrow u(a) - u_\sigma$;
18:     **end for**
19:     Insert the infoset and its action regret values $(I, r(I))$ into regret memory $M_{r,p}$;
20:     **Output:** $u_\sigma$
21: **else**
22:     $I \leftarrow s[p]$;
23:     $\sigma(s) \leftarrow$ strategy of $I$ computed using regret value $R(I, a|\theta_{r,-p})$ based on regret matching;
24:     Insert the infoset and its strategy $(I, \sigma(s))$ into strategy memory $M_{\pi,-p}$;
25:     Sample an action $a$ from distribution $\sigma(s)$;
26:     $s' = E_{\theta_e}(s, a)$;
27:     **Output:** TRAVERSE($s', p, \theta_{r,p}, \theta_{\pi,-p}, M_{r,p}, M_{\pi,-p}, E_{\theta_e}$);
28: **end if**

---

method to collect training data, and the process of external sampling is shown in Alg. 4. In this traverse function, we collect the regret training data of the traveler, and the strategy training data of other players are also gathered. After performing the traverse function several times, the regret network can be updated based on the regret memory. The above processes are repeated for $T$ times. Then the average strategy network for every player is trained based on its corresponding strategy memory. Finally, the trained average strategy networks are output as the approximate equilibrium strategy.

## F  ADDITIONAL EXPERIMENTAL RESULTS

In this section, we provide more experimental results and an ablation study. Finally, we provide the main parameters we used in our experiments.

### F.1  EXPERIMENTAL RESULTS

Here, we first verify that the performance of the model-based approach is independent of the algorithm used for computing equilibrium strategy. To this end, we perform both MB-CFR and MB-PSRO algorithms in the two-player Kuhn poker game under different sizes of offline datasets. Fig. 19 shows the results. We can find that under the same size of an offline dataset, MB-PSRO and MB-CFR achieve nearly identical results. When the size of the offline dataset increases, the performance of both algorithms becomes better. It may be caused by

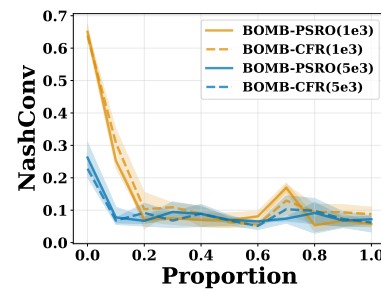

Figure 19: Results of different MB methods

the environment model being well-trained with more data. These observations indicate that the performance of the model-based algorithm is independent of the algorithm used to compute the equilibrium strategy and mainly relies on the similarity between the trained environment model and the actual environment.

### F.2  ABLATION STUDY

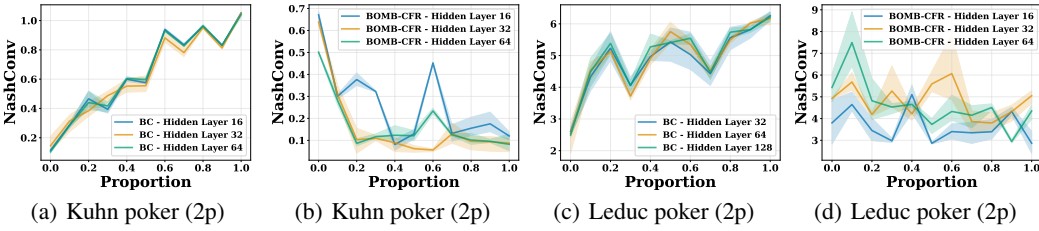

(a) Kuhn poker (2p)  (b) Kuhn poker (2p)  (c) Leduc poker (2p)  (d) Leduc poker (2p)

Figure 20: Abalation results for different hidden layer sizes.

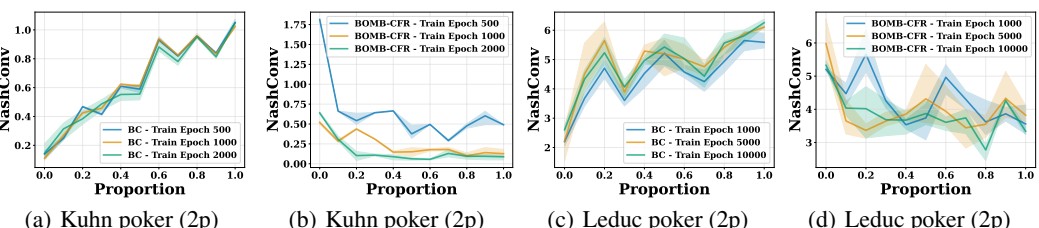

(a) Kuhn poker (2p)  (b) Kuhn poker (2p)  (c) Leduc poker (2p)  (d) Leduc poker (2p)

Figure 21: Abalation results for different train epochs.

To investigate the influence of hyperparameters, we conduct several ablation experiments on two-player Kuhn poker and Leduc poker games. We consider different model structures with various numbers of hidden layers. Specifically, for the 2-Player Kuhn poker game, we use different environment models with 16, 32, and 64 hidden layers. For the 2-Player Leduc poker game, which is a more complicated game, the numbers of hidden layers for different models are 32, 64, and 128. In addition, we train the environment models for different epochs to evaluate the robustness of our approach. Figs. 20-21 show these ablation results. We find that the number of hidden layers and the number of training epochs have little effect on the performance of the BC algorithm. These results further verify that the performance of the BC algorithm primarily depends on the quality of

the dataset. As we know, the performance of the model-based method mainly depends on the trained environment model. Since the number of the hidden layer and the number of training epochs influence the training phase of the environment model, the number of the hidden layer and the number of train epochs have a slight impact on the performance of the model-based method. As long as the size of the hidden layer and the number of training epochs can guarantee that the environment model is trained accurately, the performance of the model-based method will not be affected.

## F.3 PARAMETER SETTING

We list the parameters used to train the behavior cloning policy and environment model for all games used in our experiments in Tab. 2.

| Methods | Behavior Cloning Algorithm | | | | Environment Model Training | | | |
|---|---|---|---|---|---|---|---|---|
| Games | Kuhn Poker (2p) | | Kuhn Poker (3p) | | Kuhn Poker (2p) | | Kuhn Poker (3p) | |
| Data size | 500 | 1000 | 1000 | 5000 | 500 | 1000 | 1000 | 5000 |
| Hidden layer | 32 | 32 | 32 | 32 | 32 | 32 | 32 | 32 |
| Batch size | 32 | 32 | 32 | 32 | 32 | 32 | 32 | 32 |
| Train epoch | 1000 | 2000 | 5000 | 5000 | 1000 | 2000 | 2000 | 5000 |
| Games | Kuhn Poker (4p) | | Kuhn Poker (5p) | | Kuhn Poker (4p) | | Kuhn Poker (5p) | |
| Data size | 5000 | 20000 | 10000 | 20000 | 5000 | 20000 | 10000 | 20000 |
| Hidden layer | 64 | 64 | 64 | 64 | 64 | 64 | 64 | 64 |
| Batch size | 64 | 128 | 128 | 128 | 64 | 128 | 128 | 128 |
| Train epoch | 5000 | 5000 | 5000 | 5000 | 5000 | 5000 | 5000 | 5000 |
| Games | Leduc Poker (2p) | | Leduc Poker (3p) | | Leduc Poker (2p) | | Leduc Poker (3p) | |
| Data size | 10000 | 20000 | 10000 | 20000 | 10000 | 20000 | 10000 | 20000 |
| Hidden layer | 128 | 128 | 128 | 128 | 64 | 128 | 128 | 128 |
| Batch size | 128 | 128 | 128 | 128 | 64 | 128 | 128 | 128 |
| Train epoch | 10000 | 10000 | 10000 | 5000 | 10000 | 10000 | 10000 | 10000 |
| Games | Liar's Dice | | Phantom TTT | | Liar's Dice | | Phantom TTT | |
| Data size | 10000 | 20000 | 10000 | 20000 | 10000 | 20000 | 10000 | 20000 |
| Hidden layer | 64 | 64 | 128 | 128 | 64 | 64 | 128 | 128 |
| Batch size | 128 | 128 | 128 | 128 | 64 | 128 | 128 | 128 |
| Train epoch | 5000 | 5000 | 5000 | 5000 | 5000 | 5000 | 5000 | 5000 |

Table 2: Parameters for Behavior Cloning algorithm

