# OpenReview forum: "Offline Equilibrium Finding in Extensive-form Games: Datasets, Methods, and Analysis"
_ICLR.cc/2025/Conference — Submitted to ICLR 2025_

### Official Review · Reviewer_H5em · 2024-11-01

**Soundness:** 2
**Presentation:** 3
**Contribution:** 1
**Rating:** 3
**Confidence:** 4

**Summary:**

The paper proposes a method, called BOMB, for approximating equilibrium strategies in extensive-form games (EFGs) using offline datasets. It presents several datasets and proposes them as a benchmark for this task, while also demonstrating BOMB's ability to approximate Nash using them. The paper also presents some theoretical guarantees related to BOMB's solution quality given sufficient data and learned model expressiveness.

**Strengths:**

I think this is an important area of work that demands more attention.
* The problem setting is well-motivated; I agree with the authors that algorithms that require access to an accurate simulator or game model are difficult to apply to the real world.
* BOMB is flexible: in case the training data is from expert play, it can weigh the behavior cloning policy more, and when the data is full of many different policies, it can use the learned environment model to compute a policy.

**Weaknesses:**

Though the domain is important, I found the contributions in this paper to be insufficient to claim progress in it.

* The curated datasets do not reflect datasets that are available for the settings that Offline EF is initially motivated for. They require an accurate simulator to construct, and in the expert and learning cases, require expensive computations using the simulator. I don't think progress on these benchmarks necessarily implies progress on the problem of learning equilibria from offline data because of the data collection methods. Are there available datasets collected from human gameplay in these, or other, games that the authors could use instead?
* The assumptions significantly weaken the theoretical results in Section 4. For instance, Theorem 4.5 is essentially saying that if we have sufficiently many trajectories sampled from an approximate equilibrium profile, and a function approximator with sufficiently low error, then the strategy learned by the function approximator is also an approximate equilibrium. I don't believe this strengthens the contribution. Theorem 4.4 and 4.6 are similar.
* The approximation results for Leduc Poker are poor. A NashConv around 2 is extremely high considering the OpenSpiel implementation of CFR+ achieves an exploitability of around 0.001 in minutes on a laptop.
* The tested domains are quite small and not reflective of the realistic offline scenario that the approach is intended for. Some toy domains help, but including experiments on larger games would increase the significance of the work. Have the authors considered evaluating BOMB on larger games such as Limit Hold'em?

**Questions:**

1) Is the optimization problem in definition 2.1 that we seek to minimize over all joint strategies $\Sigma$ in the original game, or some subset of these related to $\mathcal{D}$?
2) What are the x-axes in figures 5, 6 and 7?
3) Why do the authors choose to primarily report NashConv in multi-player variants of Leduc and Kuhn when the equilibrium finding algorithm only guarantees convergence to a CCE?

**Additional Comments**

* I recommend citing Deepstack.

---

> ### Author Response · Authors · 2024-11-24
> **Reply to Reviewer H5em**
>
> Thank you for your valuable feedback. Below is our response to your question:
>
> ---
>
> **Curated Datasets**
>
> - Thank you for raising this important point. Indeed, the datasets used in our experiments are sampled from game simulators and may not fully capture the practical setting. However, these datasets were chosen intentionally to provide a controlled environment for evaluating algorithmic performance, allowing us to isolate and analyze specific factors influencing the performance of the proposed algorithm.
>
> - We agree that using datasets collected from human gameplay could offer valuable insights and strengthen the application of our methods. Unfortunately, to the best of our knowledge, publicly available datasets of human gameplay for these or similar games are currently limited. We are eager to explore real-world gameplay data in future work, should such datasets become available. Additionally, we hope that our work motivates the creation of more realistic datasets to advance Offline EF research.
>
> ---
>
> **Theoretical Results**
>
> - Thank you for your thoughtful feedback regarding the theoretical results in Section 4. We recognize that the assumptions underlying Theorems 4.4, 4.5, and 4.6 may appear strong. However, these assumptions are intended to provide a foundation base for understanding the theoretical guarantees of our method. Specifically, Theorem 4.5 demonstrates that with sufficient data and a low-error function approximator, it is possible to recover an approximate equilibrium strategy.
>
> - We agree that further relaxing these assumptions would significantly enhance the contribution, and we are actively exploring potential extensions to our theoretical framework to address more practical settings with weaker assumptions. Additionally, our empirical results aim to complement the theoretical analysis by demonstrating the effectiveness of the proposed method in scenarios where the assumptions are not strictly satisfied.
>
> ---
>
> **Results of Leduc**
>
> - We acknowledge that the NashConv values reported for Leduc Poker in our experiments are relatively high compared to the low exploitability achieved by CFR+ in the OpenSpiel implementation. This discrepancy is likely due to two key factors: the high dynamism of Leduc Poker and the poor quality of the offline dataset. Notably, even when conducting the experiments on the expert dataset, the performance of the BC algorithm is suboptimal, further indicating the low quality of the offline dataset. Additionally, the high dynamism of Leduc Poker makes it challenging to train an accurate environment model, leading to discrepancies between the trained model and the original game dynamics. These discrepancies negatively impact overall performance.
>
> ---
>
> **Small Tested Domain**
>
> - Thank you for your valuable feedback. We acknowledge that the tested domains, while effective for initial evaluation, are relatively small and may not fully capture the complexity of realistic offline scenarios. However, these domains were intentionally chosen to provide a controlled environment for understanding and validating the core principles of our approach.
>
> - We agree that evaluating our method on larger and more complex games, such as Limit Hold’em, would significantly increase the relevance and impact of our work. However, applying BOMB to larger games presents substantial computational and data-related challenges, particularly in ensuring the availability of high-quality offline datasets. Nonetheless, we are actively working toward scaling up our experiments and exploring adaptations of BOMB for larger-scale games, as we discussed in the Future Work section.
>
> ---
>
> **Reply to Q1**
>
> - In definition 2.1, we seek to minimize the gap over all joint strategies in the original game, as $\sigma^*$ in the definition represents the equilibrium strategy of the original game.
>
> ---
>
> **Reply to Q2**
>
> - The x-axis represents the proportion of data from the random dataset within the hybrid dataset. Specifically, when the proportion is 0, the hybrid dataset is equivalent to the expert dataset; conversely, when the proportion is 1, the hybrid dataset consists entirely of the random dataset. This description is provided in our paper, but we will ensure it is highlighted more explicitly for clarity.
>
> ---
>
> **Reply to Q3**
>
> - In multi-player scenarios, while these equilibrium-finding algorithms do not guarantee convergence to Nash equilibrium strategies, we report these results as part of our experiments to explore the factors influencing the performance of our model-based algorithm.  Additionally, for correlated equilibrium (CCE) solutions in multi-player cases, we also report the corresponding results in RQ4 for a more comprehensive analysis.
>
> ---
>
> We hope our response can address your concerns, and we deeply appreciate that if you could reconsider the evaluation of our paper.

---

> > ### Comment · Reviewer_H5em · 2024-11-26
> >
> > Thank you for your detailed response and your clarifications. The response doesn't change my view on the impact and significance of the theoretical contributions, and poor empirical results in Leduc are difficult to justify. I like the idea but believe a clearer indication of progress is necessary for inclusion, so I will maintain my score.

---

> > > ### Author Response · Authors · 2024-11-27
> > >
> > > Thanks for your reply.  We want to highlight that offline equilibrium finding is still at the early stage, and our paper is a preliminary attempt to address this novel and important paradigm. Our main contributions include the datasets, the algorithms, and the theoretical analysis. Compared with the work only focusing on either theory or empirical, we believe that our work is more helpful in opening new venues in game theory. Our datasets serve as the benchmarks for the following work, our algorithms serve as the baselines for researchers to improve on, and our theoretical analysis would also be helpful for theory people.
> > >
> > > Regarding the results in Leduc, paper [1] reports similar results to ours in the two-player Leduc poker game, while our results outperform those in [1] in the three-player Leduc poker game. As we previously explained, the suboptimal empirical results in Leduc may stem from the highly dynamic nature of the game, which represents an inherent challenge for offline equilibrium finding. This limitation is not unique to our work but reflects a broader difficulty within this emerging field. We believe our work represents an important step toward addressing these challenges and advancing solutions in offline equilibrium finding.
> > >
> > > We observe that this work motivates subsequent work, but unfortunately, we cannot show this due to the double-blind policy. Dear reviewers, game theory is a slowly evolving area, and we really believe that this paper can bring new insights to the community. It is important to acknowledge that no single paper can address all issues in a nascent area. We sincerely hope that the review process remains constructive, focusing on the potential of this work to bring new insights and opportunities to the community.
> > >
> > > We deeply appreciate your thoughtful feedback and sincerely hope you might reconsider the contributions of our paper.
> > >
> > > [1]Offline PSRO with Max-Min Entropy for Multi-Player Imperfect Information Games. Luo et. al.

---

> ### Author Response · Authors · 2024-12-02
> **Follow-Up on Review Discussion**
>
> Dear Reviewer H5em,
>
> Thank you once again for your valuable review and reply.
>
> As the author-reviewer discussion period is nearing its end, we would like to kindly ask if our responses have addressed your concerns. We have put a lot of effort into preparing detailed responses to your questions. We are eager to address any additional feedback or unresolved concerns you may have before the discussion period concludes.
>
> We look forward to hearing from you. Thank you for your time and consideration.
>
> Sincerely,
>
> Authors of Submission 8885

---

### Official Review · Reviewer_KpQq · 2024-11-03

**Soundness:** 2
**Presentation:** 3
**Contribution:** 2
**Rating:** 5
**Confidence:** 4

**Summary:**

The authors study offline equilibrium finding in imperfect information extensive form games. They formalize the problem and draw parallels to both the online equilibrium finding, and offline reinforcement learning (RL). They propose the BOMB algorithm, which combines two techniques from offline RL: behavior cloning and model based methods. They show that in the limit of perfect data and learning, their approach recovers the online properties. They demonstrate the applicability of their algorithm on a large number of small games.

**Strengths:**

I think offline equilibrium finding, the main focus of the paper, is interesting. The paper addresses both theoretical and empirical aspects of the problem. The paper is written well and easy to follow.

**Weaknesses:**

I believe the theoretical parts of the paper are quite weak. The theorems make strong assumptions about the size of the offline dataset, its coverage of the game, and the learning algorithm being able to perfectly learn the model of the environment. Once these are met, one can reduce the problem to the online setting, leveraging prior results. I would like to see some bound on the effectiveness of offline methods with randomly sampled datasets, assuming perfect learning. Or show how the error of the learning algorithm can affect the distance to the equilibrium similar to DeepStack (Moravcik 2017).

The BOMB algorithm seems to be a simple combination of BC and MB, with the predictor on top. I found the option of training on a smaller abstraction of a given game and scaling to a larger one quite interesting. It can maybe fit well into the offline framework, if the abstraction is created from the offline dataset.

In your experimental analysis, I would expect some results with in the “truly offline” setting, i.e. a dataset of say human play of Kuhn poker. I would expect the offline algorithms to struggle without the guaranteed coverage of the game introduced by the random policy. This would nicely illustrate the hardness of the offline setting.

Continuing in the experimental section, I find it strange that you focus on the general sum games and NE. None of the algorithms considered in the paper has guarantees to converge to a NE in a general sum game, nor do I see a good reason the BOMB should converge closer to NE. Some of the statements in the paper related to general sum games are very strong. For example, line 506-507 if true not only contradicts your earlier statements about expert strategy datasets, but also impossibility of approximating NE well in polynomial time.

There seems to be a typo on line 223.

**Questions:**

1) The predictor of BOMB takes as input the similarity between BC and MB policy. That seems strange as swapping them would not affect the distance, but it would change the optimal \alpha. Did you consider other predictors, perhaps with more context?

2) On line 409, you mention that you use the grid search method in your experiments. Doesn’t that introduce bias to your results as you pick the best \alpha in each instance?

3) Fig 1, is the strategy really a NE? It seems to me p2 would choose b2 (getting reward of 2), thus p1 would prefer a1 initially.

4) In the RQ3 section (Fig 8 and 10), you show that the BOMB can outperform MB and BC. How can that not be the case when the grid search of the BOMB predictor includes both of them and more?

---

> ### Author Response · Authors · 2024-11-24
> **Reply to Reviewer KpQq (1/2)**
>
> Thank you for your valuable feedback. Below is our response to your question:
>
> ---
>
> **Theoretical Results and Assumptions**
>
> - Thank you for your valuable feedback. We acknowledge that the theoretical results in the paper make strong assumptions, such as the dataset size, its coverage, and the learning algorithm’s ability. These assumptions were made to establish baseline guarantees for offline equilibrium finding (Offline EF) and provide a foundational understanding of the problem.
>
> - At this stage, such assumptions are an unavoidable compromise, as Offline EF introduces significant additional challenges compared to online settings, including reliance on static datasets with potentially limited coverage and imperfect learning models. By working within these assumptions, we aim to provide initial insights and a clear framework for evaluating offline methods, acknowledging their limitations.
>
> - We agree that providing bounds on the effectiveness of offline methods with randomly sampled datasets or analyzing how learning errors impact the equilibrium distance would significantly strengthen the theoretical contributions. This is a compelling suggestion and a promising direction for future work.
>
> ---
>
> **BOMB Algorithm and Abstraction**
>
> - Thank you for your insightful feedback. The BOMB algorithm combines BC and MB approaches with a predictor to determine their optimal weighting. While the overall framework may appear straightforward, its strength lies in its ability to effectively balance the strengths of BC and MB to adapt to diverse offline datasets and scenarios.
>
> - We appreciate your observation about using abstractions to scale from smaller to larger games. This is indeed an interesting direction, and we agree that creating abstractions directly from the offline dataset could fit well into the Offline EF framework. Such an approach could help address challenges related to dataset incompleteness or scalability in larger games, where direct computation might be infeasible. By extracting a meaningful abstraction from the dataset, we could potentially improve the accuracy of the environment model and the overall performance of the algorithm.
>
> - In our current work, we have not explicitly incorporated abstraction-based techniques, but we recognize their potential in the offline setting and view this as a promising avenue for future research. We appreciate your suggestion and will consider exploring how abstractions can be integrated into the BOMB framework to further enhance its applicability to large-scale games.
>
> ---
>
>
> **Truly Offline Setting in Experiments**
>
> - In our current experiments, we relied on datasets generated through random, expert, and hybrid policies to create controlled scenarios for evaluation. These allow for a systematic comparison of different methods and an understanding of how dataset quality affects performance, although they may not fully represent the challenges posed by real-world offline datasets, such as those generated from human play. As we discussed in Limitations and Future Work, we take applying the offline EF to human-play datasets as future work.
>
> ---
>
> **General Sum Games and NE**
>
> - While these equilibrium-finding algorithms do not guarantee convergence to Nash equilibrium strategies in general-sum games, we report these results as part of our experiments to explore the factors influencing the performance of our model-based algorithm.  Additionally, for correlated equilibrium (CCE) solutions in multi-player cases, we also report the corresponding results in RQ4 for a more comprehensive analysis. We appreciate your observation regarding the statements in lines 506-507. Upon review, we realize that the wording may be overly strong and could lead to confusion. We revise this section in the revision to clarify that our experimental results do not imply a general solution to approximating NE in polynomial time but rather highlight the practical effectiveness of BOMB in the specific datasets and settings tested.
>
> ---
>
> **Typo Issue**
>
> - Thanks for pointing out this. We have revised it in the revision.

---

> ### Author Response · Authors · 2024-11-24
> **Reply to Reviewer KpQq (2/2)**
>
> ---
>
> **Reply to Q1**
>
> - The current predictor design in BOMB relies on the similarity between the BC and MB policies as input, which we acknowledge may have limitations. This simplified design was chosen to provide a practical solution in a fully offline scenario.
>
> - Before finalizing this design, we also considered incorporating more contextual information into the predictor, such as the quality of the dataset or features of the environment model. However, these approaches often require additional data or computational resources, which may introduce higher demands on offline application settings. Considering these trade-offs, we opted for the similarity-based predictor as it balances feasibility and effectiveness under current resource and technical limitations. Despite its simplicity, it has shown efficacy in our experiments.
>
> - We agree that exploring more sophisticated and context-aware predictor designs is a valuable direction for future work. Such improvements could enhance the predictor’s accuracy and further expand the robustness of the BOMB framework in more complex scenarios.
>
> ---
>
> **Reply to Q2**
>
> - The purpose of using grid search in our experiments is to provide an upper-bound evaluation of the BOMB framework’s potential performance by identifying the optimal combination of BC and MB policies. We acknowledge that grid search may not reflect a fully practical offline scenario, as it requires validation data or an environment for evaluation. To address this limitation, we also proposed and evaluated a learning-based method to estimate \alpha without requiring online interactions after training. While the learning-based method performs slightly worse than grid search in some cases, it demonstrates promising results and better aligns with the fully offline setting.
>
> ---
>
> **Reply to Q3**
>
> - To answer this question, we can transfer this extensive form game into the following normal form game.
>
>     | actions | $b_1$ | $b_2$|
>     |:---:|:---:|:---:|
>     | $a_1$ |  (0, 0) |(0, 0)  |
>     | $a_2$   | (1, -1) |(-2, 2) |
>
>     We can see that action $b_2$ weakly dominates action $b_1$. When player 2 chooses action $b_2$, then player 1’s best response would be action $a_1$. Clearly, strategy profile ($a_1$, $b_2$) would be the NE strategy.
>
> ---
>
> **Reply to Q4**
>
> - Indeed, the grid search method used in BOMB allows it to explore a range of $\alpha$ values, naturally providing BOMB with the flexibility to outperform either component individually. However, the key strength of BOMB lies in its ability to effectively balance the advantages of the BC and MB approaches, as neither single method can handle all diverse cases on its own. By combining these two methods, BOMB leverages their complementary strengths, as demonstrated by the performance gains observed in our experiments. While the grid search method provides a straightforward way to optimize $\alpha$, it is the combination of BC and MB that enables BOMB to consistently achieve superior results across diverse datasets and game scenarios.

---

> ### Author Response · Authors · 2024-12-02
> **Follow-Up on Review Discussion**
>
> Dear Reviewer KpQq,
>
> Thank you once again for your valuable review.
>
> As the author-reviewer discussion period is nearing its end, we would like to kindly ask if our responses have addressed your concerns. We have put a lot of effort into preparing detailed responses to your questions. We are eager to address any additional feedback or unresolved concerns you may have before the discussion period concludes.
>
> We look forward to hearing from you. Thank you for your time and consideration.
>
> Sincerely,
>
> Authors of Submission 8885

---

### Official Review · Reviewer_rVFL · 2024-11-06

**Soundness:** 2
**Presentation:** 3
**Contribution:** 2
**Rating:** 3
**Confidence:** 5

**Summary:**

This paper proposed the problem of offline equilibrium finding, which is to find equilibrium of a multiagent game using only offline dataset. The authors proposed BOMB, which is a method that output a policy as an interpolation of a behaivor cloning policy and a model-based MARL policy. The authors also gave experimental results.

**Strengths:**

The problem of offline equilibrium finding is novel.

**Weaknesses:**

The formulation of the problem is questionable. The technical contributions are limited.

**Questions:**

The authors set an ambition goal of transferring offline RL to MARL regimes. However, my biggest concern is that there were already plenty of issues of offline RL, and adding multi-agent applications may complicate the problem more, not to mention the goal of offline MARL is questionable. Here are some detailed comments:

1. We know that offline RL have several issues. E.g., extrapolation error, distribution shift, overestimation of values. In MARL I can image these problems will even be more exaggerated. Have you observe similar issues in your studies? What is your approaches to these issues in offline MARL

2. The goal of offline equilibrium finding is even more questionable. Although the authors focus on NE or CCE which is well-defined, generally they are not unique. So how could you make sure that the state-action pairs in your dataset is essential for computing which equilibrium among all possibilities. Furthermore the offline equilibrium finding seems to highly relies on the underlying equilibrium finding algorithm you use -- whether it is CFR or PSRO. However, these algorithms themselves may introduce biases. Moreover, this paper seems to just use the vanilla version of these algorithms -- should there be any changes to address the offline setting? E.g., conservative value estimation.

3. The theoretical result looks too week. How can you be sure every state is visited. E.g., it is impossible in the game of chess or Go.

4. About BOMB How do you interpolate the two policies. Is it interpolating on the parameter space?

5. How do you apply those RL baseline in multiagent environment? Do you just let every player run this algorithm concurrently?


Some relevant papers in this direction. Could the authors compare your work with theirs:

[1] Offline PSRO with Max-Min Entropy for Multi-Player Imperfect Information Games. Luo et. al.

[2] COPSRO: An Offline Empirical Game Theoretic Method With Conservative Critic. Shao et. al.

---

> ### Author Response · Authors · 2024-11-24
> **Reply to Reviewer rVFL**
>
> Thank you for your valuable feedback. Below is our response to your question:
>
> ---
>
> **Reply to Q1**
>
> - We agree that offline MARL faces similar issues as offline RL, with these challenges being further amplified due to the added complexity of inter-agent dependencies and interactions. For instance, to mitigate the distribution shift issue, our MB approach leverages a learned environment model to simulate inter-agent dynamics and explore strategies that may not be directly represented in the dataset. To further address this, the environment model is shared among all agents, ensuring consistency and reducing potential discrepancies in the simulated interactions.
>
> ---
>
>
> **Reply to Q2**
>
> - In our model-based algorithm, the trained environment model replaces the original game simulator in the equilibrium-finding algorithms. As a result, our MB method does not exacerbate the inherent non-uniqueness problem of equilibria in these games and algorithms. While it is true that the state-action pairs in the dataset influence the training of the environment model, the primary factor affecting performance is the gap between the learned environment model and the actual game model, rather than the non-uniqueness of equilibria.
>
> - Regarding the equilibrium-finding algorithms, we intentionally use their vanilla versions in this paper to demonstrate that our MB method can seamlessly integrate online equilibrium-finding algorithms into the offline setting with minimal modifications. This approach highlights the flexibility of our framework while leaving room for future improvements to address offline-specific challenges.
>
> ---
>
> **Reply to Q3**
>
> - We acknowledge that the theoretical results rely on strong assumptions, including sufficient dataset coverage to ensure that every relevant state is visited. As you pointed out, in complex games such as chess or Go, it is practically impossible to achieve full coverage of all states due to the immense size of their state spaces.
>
> - Our theoretical framework is intended to establish a baseline understanding of the conditions under which Offline EF can succeed, even if these assumptions are idealized. In the experimental section, we demonstrate the practical performance of our approach under different offline datasets, showcasing its ability to address challenges arising from limited state coverage to some extent.
>
> ---
>
> **Reply to Q4**
>
> - We combine the two policies in the policy space. Specifically, for each state, both policies produce their respective distributions over actions. These distributions are then combined using a weighted interpolation based on the given weight, resulting in a single distribution that represents the final strategy.
>
> ---
>
> **Reply to Q5**
>
> - For the offline RL baselines, the problem is treated as an independent offline RL task for each player to compute their optimal strategy. We run the offline RL algorithms separately for each player to derive their respective optimal strategies. Once these strategies are obtained, we evaluate the gap between the resulting strategy profile and the equilibrium strategy to assess performance.
>
> ---
>
> **Relevant papers**
> - Thank you for pointing out these relevant papers. Both papers address challenges in offline multi-agent settings and propose approaches tailored to specific aspects of equilibrium finding.
>
> - Our BOMB explicitly combines behavior cloning (BC) and model-based (MB) approaches, leveraging their complementary strengths. This hybrid approach contrasts with the purely policy optimization-based frameworks in [1] and [2].
>
> - While both [1] and [2] emphasize specific response oracle and policy evaluation techniques, our work investigates the balance between BC and MB approaches to produce effective approximate equilibria.
>
> ---
>
> We hope our response can address your concerns, and we deeply appreciate that if you could reconsider the evaluation of our paper.

---

> > ### Comment · Reviewer_KpQq · 2024-11-28
> >
> > Thank you for your detailed response and your clarifications. I maintain the rating.

---

> > > ### Author Response · Authors · 2024-11-28
> > >
> > > Thanks for your reply and for taking the time to review our responses.
> > >
> > > If our responses have adequately addressed your concerns, we would be grateful if you could reconsider the evaluation of our paper. However, if there are remaining issues or concerns that you feel we have not sufficiently addressed, we kindly ask for further clarification so that we can work on solving them.
> > >
> > > Your feedback is invaluable to us, and we are committed to ensuring the contributions of this paper are clearly presented. Thank you once again for your time and effort.

---

> ### Author Response · Authors · 2024-12-02
> **Follow-Up on Review Discussion**
>
> Dear Reviewer rVFL,
>
> Thank you once again for your valuable review and reply.
>
> As the author-reviewer discussion period is nearing its end, we would like to kindly ask if our responses have addressed your concerns. We have dedicated significant effort to preparing detailed, point-by-point responses to your questions. If our responses have adequately addressed your concerns, we would be grateful if you could reconsider the evaluation of our paper. However, if there are remaining issues or concerns that you feel we have not sufficiently addressed, we kindly ask for further clarification so that we can work on resolving them before the discussion period concludes.
>
> We look forward to hearing from you. Thank you for your time and consideration.
>
> Sincerely,
>
> Authors of Submission 8885

---

### Official Review · Reviewer_PKQc · 2024-11-06

**Soundness:** 4
**Presentation:** 4
**Contribution:** 4
**Rating:** 6
**Confidence:** 3

**Summary:**

The paper investigated how to find equilibriums in extensive form games with offline datasets. There are several main contributions made by this paper. First, motivated by the fact that offline game settings are under-explored in literatures, the paper constructed diverse datasets for benchmarking the performance of offline learning. Second, the paper designed a novel framework BOMB for modeling equilibriums with offline dataset. The BOMB integrated behavioral cloning techniques into model-based method to learn the equilibriums. Third, the paper provided solid theoretical analysis of the proposed BOMB framework. Finally, extensive experiments are carried out to demonstrate the effectiveness and efficiency of BOMB.

**Strengths:**

The paper provided a very comprehensive investigation into how to learn equilibriums in multi-player games with only offline datasets. There are several strengths of the paper.

Of most interest to me is the new datasets constructed in this paper. They serve as a very good benchmark for evaluating performances of future offline RL method for learning equilibriums. This has been a critical contribution to the community.

Second, the paper provided a framework BOMB for learning equilibriums with offline datasets. The authors also derived solid theoretical analysis on their method to guarantee that approximate equilibriums can be learned with BOMB. This is a great outcome.

Finally, I appreciate that the paper conducted extensive experiments on BOMB to showcase the success of their method. These empirical results complemented their theoretical analysis and really helped readers understand the algorithm.

**Weaknesses:**

The theoretical results are not exact and explicit enough in terms of quantifying the sample complexity. For example, assumption 4.3 requires that there is sufficient amount of data, but this kind of statement is very hand wavy. I would expect a more accurate quantification of the sample complexity. I was wondering if the authors could further improve their theoretical analysis.

**Questions:**

Please provide a more accurate quantification of the sample complexity.

---

> ### Author Response · Authors · 2024-11-24
> **Reply to Reviewer PKQc**
>
> Thank you for your valuable feedback. Below is our response to your concern:
>
> ---
>
> **Sample Complexity**
>
> - We appreciate your comments on the need for more explicit theoretical analysis. In Appendix D.2, we provide a generalization bound for the training error as a function of the dataset size  $m$. Based on these results, it is possible to derive a lower bound for  $m$  by setting the training error to a target value  $\epsilon$. This analysis offers a quantification of the sample complexity within the scope of our theoretical framework.
>
> ---
>
> We hope our response can address your concerns, and we deeply appreciate that if you could reconsider the evaluation of our paper.

---

> ### Author Response · Authors · 2024-12-02
> **Follow-Up on Review Discussion**
>
> Dear Reviewer PKQc,
>
> Thank you once again for your valuable review.
>
> As the author-reviewer discussion period is nearing its end, we would like to kindly ask if our responses have addressed your concerns. We are eager to address any additional feedback or unresolved concerns you may have before the discussion period concludes.
>
> We look forward to hearing from you. Thank you for your time and consideration.
>
> Sincerely,
>
> Authors of Submission 8885

---

### Author Response · Authors · 2024-12-03

Dear Reviewers,

Thank you for your time, effort, and thoughtful feedback in reviewing our work. As the author-reviewer discussion period approaches its conclusion (11:59pm AoE on 2nd Dec), we sincerely hope that our detailed, point-by-point responses have addressed your concerns.

We have dedicated significant effort to revising our paper and preparing thorough answers to your questions. If our responses have adequately resolved your concerns, we would be deeply grateful if you could reconsider your evaluation of our submission. However, if there are any remaining issues or uncertainties that you feel we have not sufficiently addressed, we kindly ask for further clarification or suggestions. We are eager to make additional improvements before the discussion period ends to ensure our work meets your expectations.

Thank you again for your valuable feedback and for helping us improve our paper.

Sincerely,

Authors of Submission 8885

---

### Meta-Review · Area_Chair_q4cW · 2024-12-22

**Metareview:**

This paper addresses offline equilibrium finding in IIEFGs by introducing a new dataset and proposing a new algorithm, BOMB, along with accompanying analysis. The paper has received thoughtful reviews, with the main concerns summarized as follows:

- Weak Theoretical Guarantees: The theoretical results are weak, as they rely on overly strong coverage assumptions that essentially enable learning the entire model, which is theoretically unnecessary and practically infeasible.

- Dataset Limitations: It is unclear to what extent the created dataset reflects real-world offline datasets for IIEFGs. The data collection policy used in this work is likely very different from how humans play these games. Additionally, reviewers expressed concern that the tasks included are relatively simplistic, raising doubts about whether improvements on this benchmark would translate to progress in practical offline IIEFG settings.

- Practical Performance: The proposed algorithm has unsatisfactory practical performance, as highlighted by Reviewer H5em’s comments on Leduc Poker.

**Additional Comments On Reviewer Discussion:**

This paper addresses offline equilibrium finding in IIEFGs by introducing a new dataset and proposing a new algorithm, BOMB, with accompanying analysis. However, key concerns raised by reviewers remain inadequately addressed after the rebuttal phase. The theoretical guarantees rely on overly strong and impractical coverage assumptions. The dataset’s relevance to real-world offline IIEFGs is unclear, with its data collection policy differing significantly from human gameplay and its tasks being overly simplistic, raising doubts about practical applicability. Additionally, the algorithm’s practical performance is unsatisfactory, as noted in Reviewer H5em’s comments on Leduc Poker. These unresolved issues limit the work’s theoretical and practical contributions.

---

### Decision · Program_Chairs · 2025-01-22

Reject